# DERMARK: A Dynamic, Efficient and Robust Multi-bit Watermark for Large Language Models

## Abstract

As large language models (LLMs) grow more powerful, concerns over copyright infringement of LLM-generated texts have intensified. LLM watermarking has been proposed to trace unauthorized redistribution or resale of generated content by embedding identifiers within the text. Existing approaches primarily rely on one-bit watermarking, which only verifies whether a text was generated by a specific LLM. In contrast, multi-bit watermarking encodes richer information, enabling identification of the specific LLM and user involved in generated or distributed content. However, current multi-bit methods directly embed the watermark without considering its capacity, which can result in failures, especially in low-entropy texts. In this paper, we analyze that the watermark embedding follows a normal distribution. We then derive a formal inequality to optimally segment the text for watermark embedding. Building upon this, we propose DERMARK, a dynamic, efficient, and robust multi-bit watermarking method that divides the text into variable-length segments for each watermark bit during inference. Moreover, DERMARK incurs negligible overhead compared to the overall inference cost since no additional intermediate matrices are generated, and achieves robustness against text editing by minimizing watermark extraction loss. Experiments demonstrate that, compared to SOTA, on average, our method reduces the number of tokens required per embedded bit by at least 25%, reduces watermark embedding time by 50%, and maintains high robustness against text modifications and watermark erasure attacks.

## 1 Introduction

In recent years, large language models (LLMs) such as GPT-4 Achiam et al. (2023) have achieved significant advancements, excelling in various tasks such as instruction following. Training an LLM requires substantial hardware resources, vast amounts of training data, and specialized expert knowledge. Consequently, LLMs are considered valuable intellectual property (IP) of their respective owners. However, the increasing capabilities of these advanced models pose potential risks of copyright infringement, including unauthorized redistribution or resale of purchased LLM services Birch et al. (2023). Given these concerns, there is an urgent need for mechanisms to trace the distribution of LLM-generated texts, thereby protecting the IP rights of LLM owners from malicious exploitation.

LLM watermarking has emerged as a promising solution Liu et al. (2024b). By implicitly embedding watermarks into generated text, model owners can trace the downstream distribution of their outputs. Most existing watermarking methods focus on verifying whether a given text was generated by a particular LLM, known as one-bit watermarking Kirchenbauer et al. (2023). However, these approaches cannot embed user-specific identifiers and thus fail to support fine-grained attribution. To address this limitation, multi-bit watermarking has been proposed, which embeds richer information, such as binary strings Wang et al. (2024); Yoo et al. (2024), by adding biases into the LLM logits. This enables precise attribution of potentially infringing content by identifying both the LLM and the user involved in generation.

The primary factor determining the successful embedding of a multi-bit watermark in LLM-generated text is watermark capacity, defined as the maximum number of bits that can be embedded.

This capacity is directly influenced by the entropy of the text. Specifically, LLMs predict that tokens with relatively uniform probability distributions, which reflect high entropy, have a larger watermark capacity. However, existing methods ignore this property. They typically extend single-bit watermarking schemes to multi-bit scenarios based solely on the watermark length, either by hashing tokens into different segments Yoo et al. (2024) or partitioning the generated text into equal-length segments Wang et al. (2024), with each segment embedding a single bit. These segmentation strategies significantly degrade the semantics of short or low-entropy texts. In particular, when generating structured content such as code, where entropy tends to be low, watermark embedding often fails. Therefore, a crucial advancement for multi-bit watermarking in LLM-generated text is to segment the text based on its entropy dynamically. This ensures that each bit is embedded into a segment with sufficient capacity.

Designing such a multi-bit watermarking method poses three key non-trivial challenges: **(1) Lack of a principled segmentation method for LLM-generated text.** While token entropy serves as a rough proxy for estimating the watermark capacity of generated text, it remains unclear how much entropy is sufficient to reliably embed a single watermark bit. This uncertainty complicates the dynamic assignment of text segments for watermark embedding. Furthermore, due to the autoregressive nature of LLMs, where each token is generated based on the previously sampled tokens, watermark embedding inevitably perturbs the logit distribution, thereby influencing the sampling of subsequent tokens. As a result, it becomes impossible to accurately estimate the entropy of an entire segment in advance without already committing to specific token-level modifications. This entanglement creates a chicken-and-egg problem: watermark embedding decisions require prior knowledge of entropy, but embedding itself changes the entropy landscape. Effectively resolving this challenge requires a robust mechanism for aligning watermark bits with text segments in real-time during generation—a problem that remains complex and unexplored. (2) **Fragility of multi-bit watermarks to text editing.** Each bit is embedded within a specific text segment. Consequently, even minor post-editing operations, such as insertions or deletions, can disrupt segment integrity and compromise watermark extraction. This vulnerability is particularly problematic in real-world applications where LLM-generated content is often subject to downstream modifications.

In this work, we theoretically demonstrate that watermark embedding—implemented by perturbing the LLM logits—follows a normal distribution conditioned on these logits. Based on this, we derive an inequality that enables real-time assessment of whether the currently generated token sequence is sufficient to embed one watermark bit during token generation. This condition enables dynamic estimation of the required segment length for each bit. Building on this formulation, we propose DERMARK, a dynamic and efficient multi-bit watermarking method. During the embedding phase, DERMARK adaptively segments the text, using the LLM logits to guide the segmentation for each bit. Since segmentation and embedding rely solely on the logits and do not involve any intermediate matrix computations, the additional time and memory overhead is negligible compared to the overall inference cost. In the extraction phase, we apply dynamic programming to minimize the segmentation loss (inequality violations from misaligned segments) and the color loss (color imbalance in token distributions within each segment), thereby achieving robustness against text editing.

Our main contributions are threefold: (1) We formally show that watermark embedding—achieved by perturbing the LLM logits—follows a normal distribution. This enables the derivation of a closed-form expression for estimating the number of tokens required to embed each watermark bit. (2) We propose DERMARK, a dynamic, efficient, and robust multi-bit watermarking framework. It performs real-time segmentation and embedding during inference with minimal overhead. Moreover, a dynamic programming-based extraction method ensures robustness to text edits. (3) Extensive experiments demonstrate that DERMARK achieves superior efficiency, requiring on average 2.26 and 3.7 fewer tokens per embedded bit than SOTA on OPT-1.3b and LLaMA2-7b, respectively. Additionally, it maintains high robustness against text insertion and deletion, while incurring minimal inference overhead.

## 2 RELATED WORK

Since 2023, extensive efforts have been made to embed watermarks into LLM-generated text. These approaches can be broadly categorized into two paradigms: one-bit watermarking and multi-bit watermarking.

**One-bit watermarking** primarily aims to distinguish machine-generated text from human writing Kirchenbauer et al. (2023). Building upon this foundation, subsequent studies have focused on improving robustness against editing and paraphrasing Liu & Bu (2024); Hu et al. (2024); Feng et al. (2024); Liu et al. (2024a); Wan et al. (2024); Guo et al. (2024), enhancing generation quality Fu et al. (2024), or achieving task-agnostic applicability Masrani et al. (2025). However, these techniques are inherently limited to binary detection and cannot be directly extended to scenarios requiring rich information transmission, such as user attribution.

To address this limitation, **multi-bit watermarking** embeds binary strings into the text, enabling fine-grained identification across users and models. Unlike the binary nature of one-bit methods, multi-bit watermarking requires embedding a sequence of bits, which makes the segmentation strategy a central design challenge. Existing methods typically adopt one of two strategies: **Contiguous Segmentation.** This strategy allocates each bit to a block of consecutive tokens. For instance, Wang et al. (2024) proposes adding a watermark-aware loss during inference to guide bit-wise embedding into fixed-length segments. **Non-contiguous Segmentation.** This strategy distributes bits across disjoint positions. MPAC Yoo et al. (2024) determines the target bit assignment for the current token by hashing the previously generated tokens. Consequently, each bit is embedded into a logical segment composed of non-contiguous tokens spread throughout the text. Subsequently, Qu et al. (2025) enhances the robustness of this approach by incorporating error-correcting codes and uniformly discontinuous segmentation. Moreover, Zhang et al. (2024) proposes training a model to encode the watermark into the LLM-generated text. However, a key limitation shared by these methods is their reliance on fixed-pattern segmentation, which fails to consider the variable watermark capacity of different texts. Ignoring this factor can result in embedding failures, particularly in low-entropy or highly structured texts.

To address this limitation, we aim to design a dynamic multi-bit watermarking framework that, during the LLM's inference phase, dynamically estimates the segment length required for each watermark bit and adaptively segments the output text based on these estimations.

## 3 THEORETICAL ANALYSIS

This section formalizes the multi-bit watermarking problem, introduces key notations, and derives the inequalities that need to be satisfied for watermark embedding.

### 3.1 PROBLEM STATEMENT

In multi-bit watermarking, the goal is to embed a binary watermark consisting of multiple bits into the text generated by an LLM during its inference process. As previously discussed, we divide the text into segments, each dedicated to encoding a single bit. Accordingly, the embedding procedure can be conceptually divided into two steps:

**Step 1. Segmentation:** Partition the generated text into segments, each segment corresponding to one watermark bit.

**Step 2. Embedding:** Embed each bit into its corresponding segment.

Our primary focus in this work is on *Step 1-the segmentation process*, which critically determines whether the watermark embedding will succeed.

Once the text has been partitioned, we apply the one-bit watermarking technique proposed by Kirchenbauer et al. (2023) to embed each bit into its corresponding segment.

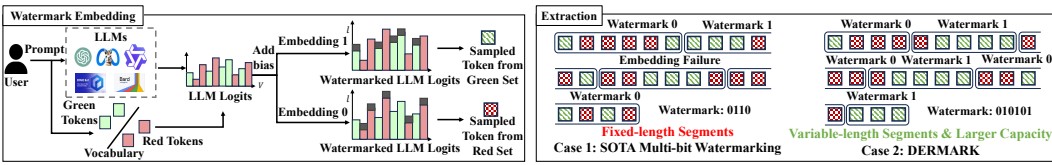

Figure 1: Multi-bit Watermarking Pipeline.

## 3.2 NOTATIONS AND PRELIMINARIES

To facilitate the understanding of our method, we provide a brief overview of the notation and processes involved in LLM inference, watermark embedding, and extraction.

During the LLM inference phase, let $\mathbf{x}^p$ denote the prefix prompt and $\mathbf{s} = \{s^{(0)}, s^{(1)}, \ldots\}$ represent the generated text, where $s^{(t)}$ denotes the $t$-th token. Let $V = \{s_1, \ldots, s_{|V|}\}$ be the vocabulary, where $s_i$ denotes the $i$-th token in the vocabulary. At each decoding step $t$, the input to the LLM consists of the prompt $\mathbf{x}^p$ and the previously generated tokens $\mathbf{s}^{:t-1} = \{s^{(0)}, \ldots, s^{(t-1)}\}$. The LLM produces a vector of logits: $\mathbf{L}(\mathbf{x}^p, \mathbf{s}^{:t-1}) = \{l_1^{(t)}, \ldots, l_{|V|}^{(t)}\}$. These logits are then passed through the softmax function to obtain the predicted token distribution: $\mathbf{P}(\mathbf{x}^p, \mathbf{s}^{:t-1}) = \{p_1^{(t)}, \ldots, p_{|V|}^{(t)}\}$, where $p_i^{(t)} = e^{l_i^{(t)}} / \sum_{j=1}^{|V|} e^{l_j^{(t)}}$. Finally, the token $s^{(t)}$ is sampled from $\mathbf{P}(\mathbf{x}^p, \mathbf{s}^{:t-1})$ using a predefined sampling strategy (e.g., probabilistic sampling).

Multi-bit watermark embedding occurs concurrently with LLM inference. Given a binary watermark $m \in \{0, 1\}^K$, each bit $m_k$ is embedded into a segment $S_k$ of consecutive tokens from $\mathbf{s}$. We apply the one-bit watermarking method Kirchenbauer et al. (2023) to embed $m_k$ into its corresponding segment $S_k$. As shown in Fig.1, the watermarking process proceeds as follows:
**1. Vocabulary Partitioning.** A random seed is used to partition the vocabulary into a *green list* $G$ and a *red list* $R$ of equal size: $G = \{s_1, \ldots, s_{|V|/2}\}$, $R = \{s_{|V|/2+1}, \ldots, s_{|V|}\}$.
**2. Logits Modification**: For bit $m_k$, if $m_k = 1$, a bias $\delta$ is added to the logits of green tokens; if $m_k = 0$, the bias is applied to red tokens: $l_i'^{(t)} = l_i^{(t)} + \delta \cdot \mathbb{I}[s_i \in C_k]$, where $C_k = G$ if $m_k = 1$, and $C_k = R$ if $m_k = 0$.
**3. Sampling.** The modified logits $\mathbf{L}'(\mathbf{x}^p, \mathbf{s}^{:t-1})$ are passed through the softmax function to obtain the watermarked distribution $\mathbf{P}'(\mathbf{x}^p, \mathbf{s}^{:t-1})$, then sample the token $s^{(t)}$. This process is repeated until all tokens in $S_k$ are generated, and the watermark bit $m_k$ is embedded into $S_k$ simultaneously.

In the watermark extraction phase, to extract the $k$-th bit $m_k'$ from a given segment $S_k$, we count the number of tokens from $G$ and $R$. If more than half of the tokens in $S_k$ belong to $G$, we decode $m_k' = 1$; otherwise, we set $m_k' = 0$. Note that in practical watermark extraction, while individual segments may occasionally yield false positives, the probability that an unwatermarked, human-written text matches a valid multi-bit watermark sequence by chance is negligible.

## 3.3 THEORETICAL DERIVATION

Building on the notation and processes introduced above, we now derive the theoretical relationship between watermark bits and the segment length required for embedding.

Since the vocabulary $V$ is partitioned into green and red lists ($G$ and $R$), we begin by analyzing the probability that the next token $s^{(t)}$ belongs to either set prior to watermarking. Let $P_G^{(t)}$ and $P_R^{(t)}$ denote the probabilities that the $t$-th token lies in $G$ or $R$, respectively:

$$P_G^{(t)} = \frac{\sum_{s_i \in G} e^{l_i^{(t)}}}{\sum_{s_i \in V} e^{l_i^{(t)}}}, \quad P_R^{(t)} = \frac{\sum_{s_i \in R} e^{l_i^{(t)}}}{\sum_{s_i \in V} e^{l_i^{(t)}}}. \tag{1}$$

Next, we analyze the probabilities after watermarking. Let $P_G'^{(t)}$ and $P_R'^{(t)}$ represent the probabilities of the token belonging to $G$ or $R$ after applying the watermarking bias. We present the following result:

**Lemma 1.** *If $m_k = 1$, $P_G'^{(t)} = \frac{e^\delta \cdot P_G^{(t)}}{e^\delta \cdot P_G^{(t)} + (1 - P_G^{(t)})}$; if $m_k = 0$, $P_R'^{(t)} = \frac{e^\delta \cdot P_R^{(t)}}{e^\delta \cdot P_R^{(t)} + (1 - P_R^{(t)})}$.*

*Proof sketch.* See Appendix D.1 for the complete proof. □

Let $P'^{(t)}$ denote the probability that the next token aligns with the watermarking intent (i.e., sampled from $G$ if $m_k = 1$, from $R$ if $m_k = 0$). We compute its expected value as:

$$\mathbb{E}[P'^{(t)}] = P(m_k = 1) \cdot P_G'^{(t)} + P(m_k = 0) \cdot P_R'^{(t)}. \tag{2}$$

To estimate the fraction of aligned tokens in a segment, let $S$ be a segment of $N$ tokens. Define $X$ as the number of tokens in $S$ that match the current bit embedding requirement, and let $T = X/N$ be the corresponding token proportion. Then, we have the following lemma:

**Lemma 2.**

$$T \sim \mathcal{N}\left(\tfrac{\mu}{N}, \tfrac{\sigma^2}{N^2}\right), \text{ where } \mu = \sum_{t=1}^{N} \mathbb{E}[P'^{(t)}], \ \sigma^2 = \sum_{t=1}^{N} (\mathbb{E}[P'^{(t)}] - \mathbb{E}^2[P'^{(t)}]). \tag{3}$$

*Proof sketch.* See Appendix D.2 for the complete proof. $\qquad\square$

To successfully embed one bit, the proportion of aligned tokens $T$ in the segment must exceed $0.5$. Since $T$ follows a normal distribution, using the significance level $\alpha$, the probability of successful embedding can be estimated as:

$$P\left(T > \frac{1}{2}\right) = \Phi\left(\frac{\mathbb{E}[T] - \frac{1}{2}}{\sqrt{\mathrm{Var}(T)}}\right) \geq 1 - \alpha \Rightarrow \Phi^{-1}(1 - \alpha) \leq \frac{\mathbb{E}[T] - \frac{1}{2}}{\sqrt{\mathrm{Var}(T)}}. \tag{4}$$

Here, $\Phi(\cdot)$ is the cumulative distribution function (CDF) of the standard normal distribution, mapping a real input to the probability that a standard normal variable does not exceed it.

We summarize this result in the following theorem:

**Theorem 1.** *When the inequality in Eq.(4) holds, the watermark bit is embedded into the segment $S$ with confidence at least $1 - \alpha$.*

The significance level $\alpha$ controls the required confidence for watermark embedding. Given a fixed $\alpha$, Theorem 1 shows that Eq.(4) enables real-time estimation of whether the current segment contains enough tokens to reliably embed the target watermark bit. Building on this result, we propose DERMARK—a dynamic, efficient, and robust method for multi-bit watermark embedding—detailed as follows.

## 4 DERMARK

### 4.1 DESIGN OVERVIEW

The overall workflow of our approach is illustrated in Fig. 2, comprising two main phases: watermark embedding and watermark extraction. During the embedding phase, LLM-generated text is dynamically segmented and watermarked based on the current bit and Eq.(4). In the extraction phase, the watermarked text is segmented using dynamic programming, jointly minimizing segmentation loss and color loss to accurately recover the embedded watermark.

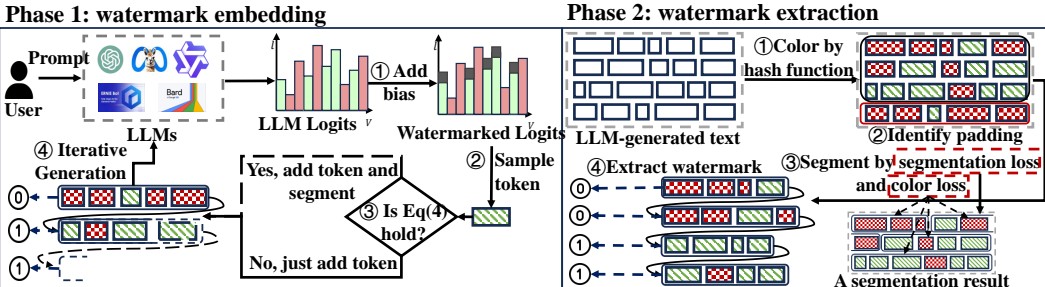

Figure 2: Workflow of DERMARK.

## 4.2 Multi-bit Watermark Embedding

During generation, as shown in phase 1 in Fig. 2, a bias is added to the logits based on the target bit. The modified logits produce a token distribution, from which we sample a token. We then use Eq.(4) to check whether the current segment can embed the bit. If the inequality holds, the segment ends and the next token starts a new segment. Otherwise, the current segment continues. This process repeats until all bits are embedded.

Nevertheless, to make this process practical for real-world applications, we propose two key enhancements:

**(1) Inequality determination.** To compute Eq.(4), we estimate $P_{k1} = P(m_k = 1)$ and $P_{k0} = P(m_k = 0)$ in Eq.(2) using the prior distribution of red and green tokens observed within the current segment:

$$P_{k1} = \frac{G_{a:b} + \lambda}{b - a + 2\lambda}, P_{k0} = \frac{R_{a:b} + \lambda}{b - a + 2\lambda}.$$  (5)

Here, $a$ and $b$ are the segment's start and current positions; $G_{a:b}$ and $R_{a:b}$ are the counts of green and red tokens. We include a smoothing hyperparameter $\lambda$ for numerical stability.

**(2) Redundant tokens.** If the generated text exceeds the required length for the watermark bits, we use the remaining tokens as padding. We flip the last bit and embed it into these remaining tokens, ensuring these tokens form a distinct final segment. This makes it easy to identify and discard padded content during extraction.

## 4.3 Watermark Extraction

Since the segmentation inequality in Eq.(4) is highly sensitive to token-level perturbations, even minor text edits can lead to different segmentation outcomes, thereby compromising watermark extraction's reliability. To enhance extraction robustness, we propose a dynamic programming-based segmentation strategy tailored for watermark recovery.

Let the segmentation of a sequence $\mathcal{S}$ be denoted as $\text{Seg}(\mathcal{S}) = \{\dots, \mathcal{S}^{(a:b)}, \dots\}$. Ideally, during segmentation, the inequality Eq.(4) is nearly tight, meaning the difference between both sides is sufficiently small. To quantify deviations from this condition, we define the segmentation loss for each segment as the squared difference between the two sides of Eq.(4):

$$\mathcal{L}_s(a, b) = \frac{(\sum_{t=a}^{b-1} \mathbb{E}[P'^{(t)}] - \frac{b-a}{2})^2}{\sum_{t=a}^{b-1} \mathbb{E}[P'^{(t)}] - \sum_{t=a}^{b-1} \mathbb{E}^2[P'^{(t)}]} - (\Phi^{-1}(1-\alpha))^2 - \epsilon_s)^2,$$

Here, $a$ and $b$ represent the start and end indices of the segment, respectively, and $\epsilon_s$ is a bias-correction hyperparameter introduced to account for systematic deviations.

Furthermore, due to the presence of the embedded watermark, each segment exhibits a pronounced color imbalance, characterized by a significantly higher proportion of either red or green tokens. To quantitatively capture this effect, we define the color loss for a segment as the normalized difference between the counts of the two token types:

$$\mathcal{L}_c(a, b) = |min(G_{a:b}, R_{a:b})/(b - a) - \epsilon_c|,$$  (6)

where $G_{a:b}$ and $R_{a:b}$ denote the number of green and red tokens, respectively, within the segment $\mathcal{S}^{(a:b)}$, and $\epsilon_c$ is a bias-correction hyperparameter accounting for systematic deviations in the expected distribution.

As a result, the watermark extraction loss is the sum of the above two losses described above:

$$\mathcal{L}(\text{Seg}) = \sum(\mathcal{L}_s(j, i) + \beta \cdot \mathcal{L}_c(j, i)),$$  (7)

where $\beta$ is a tunable parameter controlling the relative weight of the two components. To minimize this loss, we employ a dynamic programming approach to identify the optimal segmentation among all possible configurations, with a computational complexity of $\mathcal{O}(N^2)$. The complete watermark extraction workflow is illustrated in Fig. 2 and consists of the following key steps:

**1. Token Coloring.** Each token is assigned a color (red or green) based on a hash of its predecessor.
**2. Padding Identification.** Tokens associated with padding are identified by the color distribution.

Since padding embeds the inverse of the final bit of the multi-bit watermark, it can be detected based on color imbalance patterns.

**3. Segmentation.** We initialize $\epsilon_s = 0$ and $\epsilon_c = 0$ and define the loss matrix $L[k][p]$, representing the minimum loss incurred by dividing the first $p$ tokens into $k$ segments. A corresponding predecessor matrix $\text{prev}[t][b]$ tracks the start position of the segment ending at position $b$ for $t$ segments. The recurrence relation is defined as:

$$\text{if} \quad L[t-1][a] + \text{cost}[a][b] < L[t][b] : L[t][b] = L[t-1][a] + \text{cost}[a][b], \text{prev}[t][b] = a,$$

where $\text{cost}[a][b]$ denotes the loss of a single segment $\mathcal{S}^{(a:b)}$

**4. Bias Parameter Update.** Due to the effect of text editing on $\epsilon_s$ and $\epsilon_c$, we iteratively update these parameters based on the current segmentation: $\epsilon'_s(a,b) = f(\mathbb{E}[P'^{(t)}]) - (\Phi^{-1}(\alpha))^2$ and $\epsilon'_c(a,b) = \min(G_{a:b}, R_{a:b})/(b-a)$. The parameters $\epsilon_s$ and $\epsilon_c$ are updated to the min values of $\epsilon'_s(a,b)$ and $\epsilon'_c(a,b)$ across all segments.

**5. Iterate Until Convergence.** Steps 3 and 4 are repeated until both $\epsilon_s$ and $\epsilon_c$ converge to stable values.

**6. Extract the Watermark from Segments.** The watermark bit $m'_k$ is extracted from each segment based on the relative proportion of red and green tokens. The overall watermark detection rate is computed as:

$$d_r = 1 - \sum_k \frac{|m_k - m'_k|}{K}. \tag{8}$$

It is important to note that the watermark detection rate reported in this work is mathematically equivalent to the bit match rate, defined as the proportion of matching bits between the extracted watermark and the original watermark.

## 5 EXPERIMENTS

In this section, we present extensive experiments to demonstrate the superior performance of DER-MARK in terms of watermark capacity, impact on text quality, efficiency and robustness.

### 5.1 EXPERIMENTAL SETUP

All experiments are conducted using two widely adopted LLMs: `OPT-1.3b` Zhang et al. (2022) and `LLaMA-2-7b` Touvron et al. (2023). For evaluation, we follow prior work by leveraging the news-like subset of the C4 dataset Raffel et al. (2020) as input prompts and set the watermark strength $\delta$ to 1. During inference, prompts are randomly sampled from this subset and truncated to the first 100 tokens. Token generation is performed via multinomial sampling, with a repetition penalty of 1.5 to encourage lexical diversity and mitigate mode collapse. $\alpha$ is set within the range $[0.8, 0.99]$, $\beta$ is fixed at 34, and $\lambda$ is set to $\alpha \cdot (\Phi^{-1}(\alpha))^2$. We compare our method against two baselines: Balance-Marking Wang et al. (2024), using `GPT-2` as its auxiliary model; MPAC Yoo et al. (2024), where we set $\gamma = 0.5$ to ensure consistency with our experimental setting. All experiments were performed multiple times on a server running Ubuntu 24.04.2 LTS and averages were calculated as results. The hardware environment consists of an Intel Xeon Gold 6426Y CPU and four NVIDIA A100-SXM4-80GB GPUs. Python 3.12.8 is used as the primary development environment.

### 5.2 WATERMARK CAPACITY

To demonstrate the superior watermark capacity of DERMARK, we first evaluate its performance against Balance-Marking and MPAC on the evaluation dataset. Watermark capacity is assessed by varying the value of $\alpha$ in DERMARK. Specifically, for each $\alpha$, we prompt the LLM to generate 500 texts and compute two key metrics: the average watermark detection rate and the average number of tokens required to embed each bit. For the baselines, we apply their respective watermarking schemes to the generated texts and evaluate them using the same metrics.

As shown in Fig. 3a, DERMARK consistently outperforms both baselines. It saves approximately 2.26 to 3.7 tokens per bit against Balance-Marking, and widens the gap to over 5 tokens per bit against MPAC. Furthermore, MPAC requires a substantial length of 30 tokens per bit to attain a 90% detection rate, and Balance-Marking fails to reach it entirely on `LLaMA-2-7b`, DERMARK achieves over 95% accuracy with minimal token consumption.

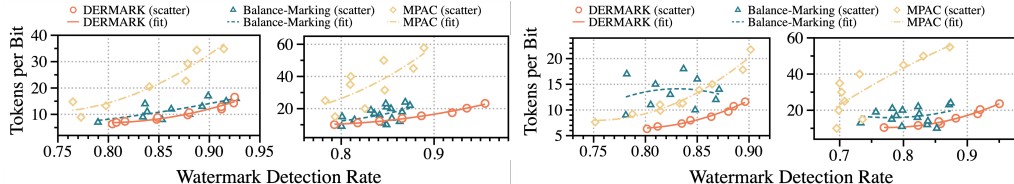

(a) Capacity comparison across total dataset on `OPT-1.3b` (left) and `LLaMA-2-7b` (right).

(b) Capacity comparison across low-entropy dataset on `OPT-1.3b` (left) and `LLaMA-2-7b` (right).

Figure 3: Capacity Comparison. Each scatter point in figure is measured over 500 samples under different preset $\alpha$ values.

To further highlight DERMARK's capacity, we construct a "low-entropy" dataset comprising the worst-performing 25% of evaluation samples (based on the watermark detection rate under the Balance-Marking scheme). We then evaluate all methods on this challenging subset. As shown in Fig. 3b, DERMARK significantly outperforms the baselines. In terms of capacity, it requires at least 5 fewer tokens per bit compared to Balance-Marking to achieve the same detection rate. Notably, this efficiency gap widens against MPAC, where DERMARK saves at least 10 tokens per bit on both `OPT-1.3b` and `LLaMA-2-7b`. Furthermore, DERMARK yields more stable results with fewer outliers.

Finally, we evaluate the watermark capacity under varying watermark strength $\delta$. As shown in Table 1, DERMARK consistently outperforms the Balance-Marking and MPAC across all $\delta$ values on both `OPT-1.3b` and `LLaMA-2-7b`.

Table 1: Number of tokens required per watermark bit to achieve a 90% watermark detection rate under varying watermark strengths $\delta$. "$-$" indicates failure to reach the target accuracy.

| Model | Method \ $\delta$ | 0.5 | 0.8 | 1.0 | 1.2 | 1.5 | 1.8 | 2.0 |
|---|---|---|---|---|---|---|---|---|
| `OPT-1.3b` | DERMARK | 35.12 | 12.59 | 10.78 | 8.05 | 5.12 | 3.48 | 2.97 |
| | Balance-Marking | $-$ | 16.00 | 12.80 | 9.31 | 7.13 | 4.57 | 3.32 |
| | MPAC | 83.50 | 33.46 | 27.10 | 20.27 | 18.74 | 9.93 | 8.67 |
| `LLaMA-2-7b` | DERMARK | 45.93 | 22.35 | 16.25 | 9.50 | 7.37 | 5.33 | 4.10 |
| | Balance-Marking | $-$ | $-$ | 41.89 | 24.12 | 17.39 | 14.06 | 6.13 |
| | MPAC | $-$ | 119.95 | 74.96 | 44.98 | 27.21 | 20.00 | 14.86 |

## 5.3 IMPACT OF WATERMARKING ON TEXT QUALITY

We investigate how watermark strength affects the quality of generated text by varying the value of $\delta$. A total of 500 prompts are randomly sampled from the evaluation set. For each $\delta$ value, we use `OPT-1.3b` and `LLaMA-2-7b` to generate responses and evaluate the resulting texts using the Perplexity (PPL) metric computed by `LLaMA-2-13B`, which serves as a proxy model to assess fluency. Lower PPL values indicate higher textual quality. Table 2 reports the average PPL across all prompts for each watermarking method under different $\delta$ settings. All methods exhibit similar trends in PPL as $\delta$ varies. Overall, our approach achieves comparable text quality to the baseline.

## 5.4 EFFICIENCY ANALYSES

We evaluate the efficiency of Balance-Marking, MPAC, and DERMARK in terms of both text generation (watermark embedding) and watermark extraction. All methods are tested with a fixed prompt length of 100 tokens. Specifically, we compare the average time overhead required for each method to achieve a 90% watermark detection rate.

The efficiency results are as shown in Fig.4. In the embedding phase, both DERMARK and MPAC incur negligible additional time overhead, maintaining inference speeds consistent with standard

Table 2: Perplexity under varying watermark strengths $\delta$; $\delta = 0$ indicates original model outputs.

| Model | Method \ $\delta$ | 0 | 0.5 | 0.8 | 1.0 | 1.2 | 1.5 | 1.8 | 2.0 |
|---|---|---|---|---|---|---|---|---|---|
| OPT-1.3b | DERMARK | 6.21 | 6.56 | 6.90 | 7.21 | 8.12 | 7.92 | 7.83 | 8.85 |
| | Balance-Marking | 6.21 | 6.69 | 7.47 | 6.13 | 6.69 | 7.47 | 8.75 | 9.04 |
| | MPAC | 6.21 | 7.12 | 7.06 | 7.43 | 7.50 | 7.11 | 9.00 | 9.82 |
| LLaMA-2-7b | DERMARK | 2.07 | 3.03 | 3.17 | 4.92 | 4.23 | 4.74 | 5.64 | 5.56 |
| | Balance-Marking | 2.07 | 3.34 | 3.08 | 5.19 | 4.74 | 4.92 | 5.13 | 6.59 |
| | MPAC | 2.07 | 3.20 | 2.26 | 3.09 | 4.95 | 3.36 | 5.26 | 6.13 |

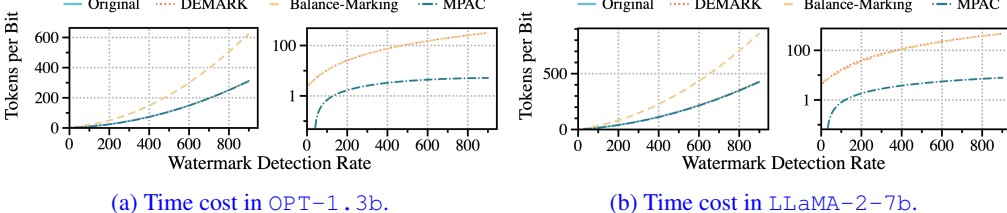

(a) Time cost in OPT-1.3b.  (b) Time cost in LLaMA-2-7b.

Figure 4: Comparison of time cost in text generation (left) and watermark extraction (right) on OPT-1.3b and LLaMA-2-7b.

model inference. In contrast, Balance-Marking exhibits significantly higher latency, approximately doubling the inference time. In the extraction phase, MPAC achieves significantly lower extraction latency, attributed to its linear $O(N)$ computational complexity. Conversely, DERMARK and Balance-Marking show comparable extraction times, governed by an $O(N^2)$ complexity.

Overall, DERMARK incurs negligible embedding overhead to ensure an optimal user experience during real-time LLM inference services. Since watermark extraction is a low-frequency, on-demand operation performed only during copyright verification, we consider DERMARK's extraction cost to be a highly acceptable trade-off.

## 5.5 ROBUSTNESS

We evaluate the robustness of DERMARK against insertion and deletion, showing that it maintains high detection accuracy under such conditions.

**Insertion Attacks.** We assess robustness to insertion by adding 5% and 10% random tokens to the generated texts. As shown in Fig. 5a and 5b, DERMARK consistently achieves higher watermark detection rates with fewer tokens per embedded bit compared to Balance-Marking and MPCA. The scatter plots and fitted trend lines indicate that DERMARK's performance curve lies below that of the baselines, underscoring its resilience to insertion noise.

**Deletion Attacks.** To evaluate deletion robustness, we randomly remove 5% and 10% of tokens. As shown in Fig. 5c and 5d, DERMARK outperforms baselines, with lower tokens-per-bit and higher detection accuracy, especially at higher thresholds.

**Substitution Attacks.** To evaluate substitution robustness, we randomly remove 5% and 10% of tokens. As shown in Fig. 5e and 5f, all methods exhibit comparable robustness profiles with no significant performance gaps. This confirms that DERMARK maintains a standard level of resilience against token replacement, performing on par with strong baselines.

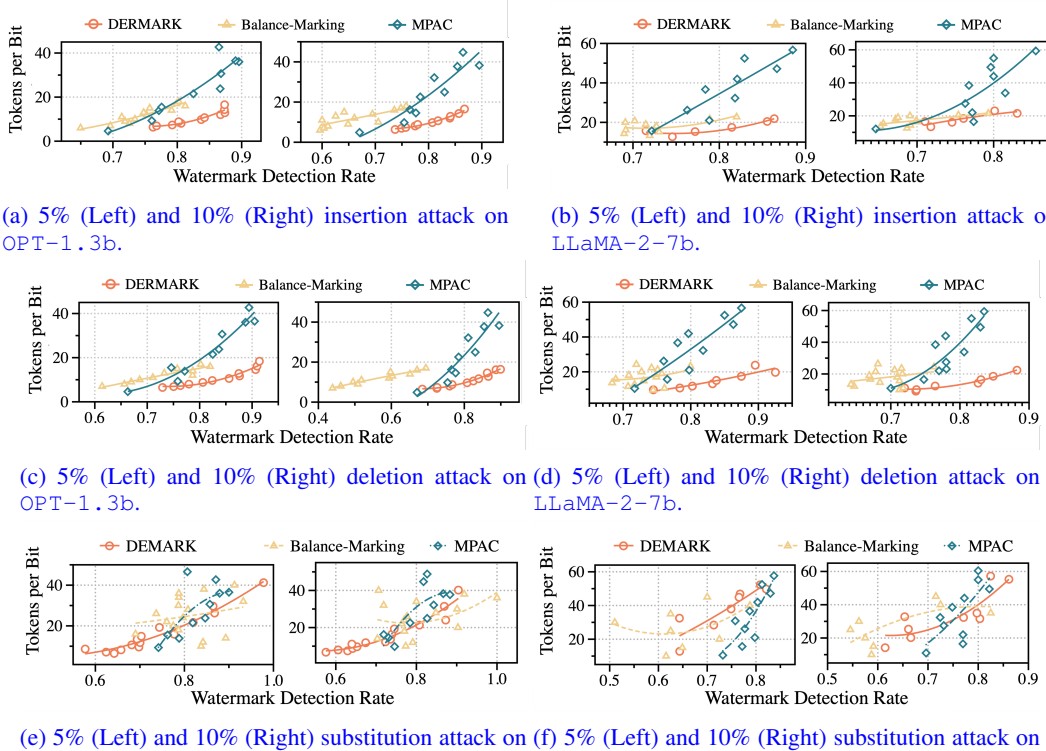

(a) 5% (Left) and 10% (Right) insertion attack on `OPT-1.3b`.

(b) 5% (Left) and 10% (Right) insertion attack on `LLaMA-2-7b`.

(c) 5% (Left) and 10% (Right) deletion attack on `OPT-1.3b`.

(d) 5% (Left) and 10% (Right) deletion attack on `LLaMA-2-7b`.

(e) 5% (Left) and 10% (Right) substitution attack on `OPT-1.3b`.

(f) 5% (Left) and 10% (Right) substitution attack on `LLaMA-2-7b`.

Figure 5: Robustness comparison. Each scatter point in the figures is measured over 500 samples under different preset $\alpha$ values.

## 6    CONCLUSION

In this work, we investigate strategies for dynamically assigning text segments to individual watermark bits and propose DERMARK, a lightweight framework that satisfies dynamic embedding requirements with negligible overhead. Extensive experiments on watermark capacity, efficiency, and robustness demonstrate that DERMARK provides a practical and effective solution for multi-bit watermark embedding.

## 7    ETHICS STATEMENT

In developing DERMARK to advance the embedding of multi-bit watermarking into LLMs, we have taken great care in our data practices. All datasets and models used in this study were derived from well-recognized previously published works, ensuring that they do not contain personally identifiable information. In addition, the evaluation benchmarks we employ are consistent with those established in previous studies, effectively eliminating the risk of privacy violations or data breaches.

## 8    REPRODUCIBILITY STATEMENT

To ensure the reproducibility of our work, we provide detailed descriptions of the DERMARK algorithm, including its segmentation process and multi-bit watermark embedding scheme, within the main text. All theoretical claims and derivations are fully explained in the appendix. Experimental settings, dataset preprocessing steps, and evaluation protocols are clearly documented in the main text and appendix, enabling other researchers to reproduce the results reported in this paper.

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

## A LIMITATIONS

The core mechanism of DERMARK lies in its segmentation process rather than the embedding step. Within each segment, a one-bit watermarking scheme is employed, allowing DERMARK to leverage existing one-bit watermarking techniques originally developed to enhance robustness against watermark-erasure attacks. Importantly, DERMARK imposes no restrictions on the choice of watermarking strategy and is, in principle, compatible with both semantically neutral watermarking Hu et al. (2024) and flexible watermarking Wang et al. (2025). While this design suggests a potential improvement in resilience against paraphrasing attacks, we note that this observation remains conceptual rather than empirically validated. A systematic evaluation of this aspect is left to future work.

Despite these strengths, our method exhibits two primary limitations. First, it is applicable only to autoregressive large language models, with watermarks embedded exclusively in textual outputs. It is not designed for models that generate non-textual content, such as those used in text-to-image tasks. Second, the method lacks robustness against substantial text modifications, including large-scale reordering, insertion, deletion, or extensive rewriting. This limitation is not unique to our approach but is inherent to all multi-bit watermarking schemes for large language models, as it originates from the underlying KGW watermarking mechanism. In particular, because multiple bits are dispersed across the text, they are inherently susceptible to flips under aggressive editing, which significantly degrades watermark extraction.

Finally, we re-emphasize our key contribution: the dynamic segmentation mechanism. This design effectively harnesses the text's capacity to carry watermarks, enabling high-confidence multi-bit embedding while minimally affecting semantic fidelity. Furthermore, in terms of robustness, our method demonstrates substantial improvements over the baseline across diverse adversarial scenarios.

## B PERFORMANCE ANALYSIS

**Larger capacity**. Unlike prior methods that allocate a fixed number of tokens per bit, DERMARK introduces a dynamic segmentation strategy that adapts to the token-level watermark capacity, leading to larger capacity. This adaptive segmentation mechanism also improves performance on longer texts, effectively balancing watermark embedding strength and semantic preservation.
**Architecture-agnostic**. It only modifies the logits, making it fully compatible with autoregressive

LLMs (e.g., GPT, Qwen) without requiring retraining or architecture changes.

**Efficient and deployment-friendly**. Since DERMARK operates only through a lightweight post-processing layer, it avoids intermediate matrix computations, thus achieving watermark embedding and segmentation in linear time with the complexity of $\mathcal{O}(N)$. Furthermore, the embedding process is compatible with `transformers.watermarking_config` interface, facilitating seamless integration into existing inference pipelines with minimal engineering effort.

**Robust against text editing**. Constraining segmentation through a loss-based formulation can effectively reduce the impact of text editing on extraction. The introduction of relaxation terms $\epsilon$ helps to accommodate minor token modifications, thereby reducing sensitivity in evaluating the inequality condition in Eq.(4), and maintaining reliable extraction even under perturbations.

## C  BASELINE SELECTION

We acknowledge that the baseline method used in our experiments is not the most recent work Yoo et al. (2024) in the field. In contrast, we select the SOTA method Balance-Marking as the baseline. This choice was made intentionally to ensure a clear and fair comparison focused on core methodological differences.

Here we explain why we choose Balance-Marking as the baseline instead of *MPAC* Yoo et al. (2024):

**(1) Methodological Innovation in Balance-Marking.** Balance-Marking introduces a principled and effective approach to constructing watermark vocabularies. It selects top-ranked tokens under an auxiliary model until a predefined cumulative probability threshold (e.g., 0.5) is reached, forming the green list. This design explicitly leverages token likelihoods in a probabilistic manner, thereby improving both the efficiency and robustness of watermark embedding. In contrast, *MPAC* merely modifies the position allocation strategy of the existing single-bit method *KGW* when extended to the multi-bit setting. It does not propose any novel mechanism for vocabulary selection or multi-bit encoding. As such, it lacks methodological advancement beyond the use of positional cues.

**(2) Incomplete and Biased Use of Balance-Marking in *MPAC*.** Although *MPAC* includes Balance-Marking as a baseline in its experiments, it does not faithfully reproduce the original method. Instead, it employs a simplified variant, citing incompatibility between the tokenizer of the main model and that of the auxiliary model required by Balance-Marking. However, this issue was explicitly addressed in the original Balance-Marking paper, which proposes a straightforward solution: using the target model itself as the auxiliary model to ensure tokenizer compatibility. By disregarding this solution, *MPAC* ends up using a weakened version of Balance-Marking, resulting in a baseline comparison that underrepresents its true performance.

**(3) Misleading Performance Gains via Aggressive Parameter Tuning.** In its experiments, *MPAC* sets the watermark strength parameter $\delta$ to 2, which is significantly higher than the widely adopted default value of $\delta = 1$ in most prior watermarking studies. It is important to note that in *KGW*—the base method upon which *MPAC* builds—the watermarking strength scales exponentially with $\delta$, i.e., the intensity is proportional to $e^{\delta}$. Thus, increasing $\delta$ from 1 to 2 results in approximately a $7.39\times$ increase in watermark strength, artificially boosting detectability.

However, this apparent gain comes at the cost of text quality and practicality, making such parameter settings unsuitable for fair evaluation. For instance, under $\delta = 2$, *MPAC* achieves a watermark detection accuracy of 0.899 when embedding 24 bits in 250 tokens, equivalent to 10.42 tokens per bit. In contrast, our *Balance-Marking* method reaches 0.900 accuracy at just 6.13 tokens per bit, and our proposed *DERMARK* further improves this to 0.900 accuracy with only 4.10 tokens per bit—demonstrating markedly better capacity-efficiency tradeoffs under standard $\delta = 1$ settings.

Moreover, since *MPAC* only modifies the position allocation mechanism without enhancing the vocabulary construction or bit encoding strategies, its ability to simultaneously improve watermark capacity and generation quality is inherently limited. Consequently, the observed performance gains are largely attributable to aggressive hyperparameter tuning rather than substantive methodological improvements.

In summary, we select Balance-Marking as our baseline due to its clear methodological contributions, faithful and complete implementation, and adherence to fair and standardized evaluation protocols. While it is not the most recent work in the field, it remains the most effective among existing

multi-bit watermarking methods in terms of robustness and detection performance. This choice ensures a more meaningful and rigorous comparison within the multi-bit watermarking landscape.

# D   DETAILED PROOFS

## D.1   PROOF OF LEMMA 1

*Proof.* Taking $m_k = 1$ as an example.

For $s_i \in G$, the predicted probability of $s_i$ is:

$$p_{Gi}^{\prime(t)}(s_i \mid \mathbf{x}^{prompt}, \mathbf{s}^{:t-1}) = \frac{e^{l_i^{(t)} + \delta}}{\sum_{s_i \in G} e^{l_j^{(t)} + \delta} + \sum_{s_i \in R} e^{l_j^{(t)}}}.$$

For $s_i \in R$, the predicted probability of $s_i$ is:

$$p_{Ri}^{\prime(t)}(s_i \mid \mathbf{x}^{prompt}, \mathbf{s}^{:t-1}) = \frac{e^{l_i^{(t)}}}{\sum_{s_i \in G} e^{l_j^{(t)} + \delta} + \sum_{s_i \in R} e^{l_j^{(t)}}}.$$

It is easy to see that $P_G^{\prime(t)}$ is the sum of $p_{Gi}^{\prime(t)}$:

$$P_G^{\prime(t)} = \sum_{s_i \in G} p_{Gi}^{\prime(t)}(s_i \mid \mathbf{x}^{prompt}, s^{:t-1})$$

$$\Rightarrow P_G^{\prime(t)} = \frac{e^{\delta} \sum_{s_i \in G} e^{l_i^{(t)}}}{e^{\delta} \sum_{s_i \in G} e^{l_j^{(t)}} + \sum_{s_i \in R} e^{l_j^{(t)}}} = \frac{e^{\delta} \cdot P_G^{(t)}}{e^{\delta} \cdot P_G^{(t)} + (1 - P_G^{(t)})}.$$

$\square$

## D.2   PROOF OF LEMMA 2

*Proof.* As defined above, $X$ counts the number of aligned tokens in the segment $S$, where each $P^{\prime(t)} \in (0, 1)$ indicates whether the $t$-th token matches the bit embedding requirement. Since token generation is independent and the target list is re-sampled at each time step, the random variables $\{P^{\prime(t)}\}_{t=1}^N$ are mutually independent but not identically distributed.

Therefore, $X$ follows a **Poisson binomial distribution**, i.e., the sum of independent (but non-identical) Bernoulli variables.

From the properties of the Poisson binomial distribution, we have:

$$\mathbb{E}[X] = \sum_{t=1}^N \mathbb{E}[P^{\prime(t)}], \mathrm{Var}(X) = \sum_{t=1}^N \mathbb{E}[P^{\prime(t)}] \left(1 - \mathbb{E}[P^{\prime(t)}]\right).$$

By the **Central Limit Theorem** (CLT) for Poisson binomial distributions (cf. Tang & Tang (2023)), as $N$ becomes large, the distribution of $X$ can be approximated by a normal distribution:

$$X \sim \mathcal{N}(\mu, \sigma^2), \text{ where } \mu = \mathbb{E}[X], \sigma^2 = \mathrm{Var}(X).$$

Since $T$ is a linear transformation of $X$, it also approximately follows a normal distribution:

$$T \sim \mathcal{N}\left(\frac{\mu}{N}, \frac{\sigma^2}{N^2}\right), \text{ where } \mu = \sum_{t=1}^N \mathbb{E}[P^{\prime(t)}], \sigma^2 = \sum_{t=1}^N (\mathbb{E}[P^{\prime(t)}] - \mathbb{E}^2[P^{\prime(t)}]).$$

$\square$

# E   ALGORITHM

## E.1   WATERMARKED TEXT GENERATION

To embed a watermark into a sequence of tokens, we propose a segmentation approach founded on the following inequality:

---

**Algorithm 1** Text Generation with DERMARK

---

**Input:** Prompt: $\mathbf{x}^{prompt}$, Bit Error Rate: $\alpha$, Binary message: $\mathcal{M}$, Maximum generation length: L
**Output:** A token string that carries $\mathcal{M}$: $\mathcal{S}$
1: Append the $(1 - \mathcal{M}[k-1])$ token to $\mathcal{M}$
2: $i \leftarrow 0$
3: $P \leftarrow \{\}$
4: **for** $t = 0, 1, \ldots, L$ do **do**
5:     $k \leftarrow \mathcal{M}[i]$
6:     Apply the LLM to prior tokens $\{\mathbf{x}^{prompt}, s^{(0)}, \ldots, s^{(t-1)}\}$ to get a logit vector $\mathbf{L}(\mathbf{x}^{prompt}, \mathbf{s}^{:t})$
7:     Using $s^{(t-1)}$to seed a randomly partition the vocabulary into two identically sized sets: $G$, $R$.
8:     Based on bit $k$, choose either $G$ or $R$ to apply gain enhancement.
9:     Using soft watermark method, watermark the token sample the next token, $s^{(t)}$
10:     Substitute $G$, $R$, $\alpha$, and $l^{(t)}$ into Eq.equation 9 to compute $\mathbb{E}[P'^{(t)}]$.
11:     Add $\mathbb{E}[P'^{(t)}]$ to set $P$
12:     **if** equation 9is satisfied on $P$ **then**
13:         $P \leftarrow \{\}$
14:         **if** $i \leq k$ **then**
15:             $i \leftarrow i + 1$
16:         **end if**
17:     **end if**
18: **end for**
19: **return**  Outputs

---

$$\Phi^{-1}(1 - \alpha) \leq \frac{\frac{1}{2} - \mathbb{E}(T)}{\sqrt{\text{Var}(T)}}. \tag{9}$$

This inequality serves as the foundation for identifying segment boundaries during the watermark extraction process. Below, we present Algorithm 1, which details the watermark extraction procedure. The algorithm can be divided into several logical groups:

**Input and Output:**   The input includes a prompt $\mathbf{x}^{prompt}$, a binary message $\mathcal{M}$, a predefined Bit Error Rate (BER) $\alpha$, and a maximum generation length $L$. The output is a generated token sequence $\mathcal{S} = \{s^{(0)}, \ldots, s^{(L)}\}$ that encodes the binary message $\mathcal{M}$ using dynamic segment marking.

**Initialization (Lines 1-3):**   An empty set $P$ is initialized to temporarily store the computed expectations $\mathbb{E}[P'^{(t)}]$. The index counter $i$ is set to 0, which tracks the current position in the binary message $\mathcal{M}$.

**Token Processing (Lines 5-11):**   For each step $t$ in the token generation process:

- Retrieve the current bit $k = \mathcal{M}[i]$ (Line 5).
- Compute the logit vector $\mathbf{L}(\mathbf{x}^{prompt}, \mathbf{s}^{:t})$ based on the sequence of prior tokens, using the large language model (Line 6).
- Partition the vocabulary into two equal subsets $G$ and $R$, using the previous token $s^{(t-1)}$ as a random seed (Line 7).
- Apply gain enhancement to $G$ or $R$ depending on the value of $k$ to modulate the probability distribution (Line 8).
- Use a soft watermarking method to sample the next token $s^{(t)}$ from the adjusted probability distribution (Line 9).
- Compute $\mathbb{E}[P'^{(t)}]$ using $G$, $R$, $\alpha$, and the current logit vector, and append it to the set $P$ (Line 10).

---

**Algorithm 2** Robust Segmentation

---

**Input:** A token string that carries $\mathcal{M}$: $\mathcal{S} = \{s^{(0)}, \dots, s^{(L)}\}$, Bit Error Rate: $\alpha$ , bit number:k
**Output:** Segmentation for the Token string:Segments
 1: Calculate the membership of each token $s^{(t)}$ in either $V_0$ or $V_1$, and store the result in bit
 2: Calculate the $P_0^{(t)}$ for each token $s^{(t)}$, and store the result in $P$
 3: Initialize two 2D arrays $\mathcal{L}$ and prev of size k $\times$ $(L+1)$
 4: **for** $p = 1, 2, \dots, $k do **do**
 5:      **for** $q = 1, 2, \dots, $L $+1$ do **do**
 6:         $\mathcal{L}[p][q] \leftarrow [\infty] * (L+1)$
 7:      **end for**
 8: **end for**
 9: **for** $t = 1, 2, \dots, k$ do **do**
10:      **for** $b = 1, 2, \dots, L$ do **do**
11:         **for** $a = 0, 1, \dots, b-1$ do **do**
12:            compute $\mathrm{cost}[a][b]$
13:            **if** $\mathcal{L}[t-1][a] + \mathrm{cost}[a][b] < \mathcal{L}[t][b]$ **then**
14:              $\mathcal{L}[t][b] \leftarrow \mathcal{L}[t-1][a] + \mathrm{cost}[a][b]$
15:              $\mathrm{prev}[t][b] \leftarrow a$
16:            **end if**
17:         **end for**
18:      **end for**
19: **end for**
20: Segments $\leftarrow []$
21: current $\leftarrow n$
22: **while** current $> 0$ **do**
23:      start $\leftarrow$ Seg[current]
24:      Append $(\mathrm{start}, \mathrm{current})$ to Segments
25:      current $\leftarrow$ start
26: **end while**
27: **return** Segments

---

**Segmentation and Message Update (Lines 12-17):**

- Periodically check whether the consistency condition in Eq.equation 9 is satisfied for the accumulated probabilities in $P$ (Line 13).

- If the condition is satisfied:

  - Compare the counters $G$ and $R$ to determine the corresponding bit for the segment, and update the binary message $\mathcal{M}'$ (Lines 14-16).
  - Increment the index counter $i$ to proceed to the next bit in $\mathcal{M}$ (Line 17).

- Reset $P$, $G$, and $R$ for processing the next segment (Lines 18-19).

E.2    WATERMARK EXTRACTION

Algorithm 2 outlines a method for reconstructing text into $k$ segments during watermark embedding, with padding tokens removed, through the minimization of a loss function.

**Input and Output:**    The input is a token string $\mathcal{S} = \{s^{(0)}, \dots, s^{(L)}\}$ that carries the binary message $\mathcal{M}$, a predefined Bit Error Rate (BER) $\alpha$, and the number of segments $k$. The output is the segmentation of the token string, represented as a list of segment boundaries.

**Preprocessing (Lines 1-3):**

- Compute the membership of each token $s^{(t)}$ in either $V_0$ or $V_1$, and store the results as binary values in the array 'bit' (Line 3).

- Compute $P_0^{(t)}$, the probabilities associated with $s^{(t)}$, and store these values in the array $P$ (Line 4).

- Initialize two 2D arrays: $\mathcal{L}$ and prev of size $k \times (L + 1)$ to store the cumulative loss for each segmentation scenario and prefix information

**Initialization of Loss Table (Lines 4-8):** Set all entries in $\mathcal{L}$ to infinity, ensuring that only valid segmentation paths are selected during subsequent calculations (Lines 6-8).

**Dynamic Programming for Segmentation (Lines 9-19):**

- For each segment index $t$ and each token position $i$, evaluate all possible preceding segment boundaries $j$ (Lines 10-12).

- Compute cost$[a][b]$, which represent the loss for the segment $(a, b)$ (Lines 13-14).

- If the total loss for the current segmentation path, $\mathcal{L}[t-1][a] + \text{cost}[a][b]$, is smaller than the existing loss at $\mathcal{L}[t][b]$, update $\mathcal{L}[t][b]$ and record the prefix position $j$ in prev$[t][b]$ (Lines 15-18).

**Backtracking to Extract Segments (Lines 20-26):**

- Start from the last position of the token string and iteratively trace back using the boundary indices stored in prev$[t][b]$ to recover all segment boundaries (Lines 23-26).

- Append each segment as a pair of start and end indices $(\text{start}, \text{current})$ to the 'Segments' list (Line 25).

**Termination (Line 27):** Return the final list of segment boundaries, 'Segments', which represents the robust segmentation of the token string based on the minimum cumulative loss.

# F  DERMARK IN PRACTICAL SCENARIOS

## F.1  TEXT LENGTH

DERMARK is capable of efficiently embedding multi-bit watermarks into text generated by large language models (LLMs). However, in practical use, the text length and watermark capacity may not ideally align with the embedding requirements of DERMARK. Therefore, in this section, we discuss two extreme cases: excessively short and excessively long text lengths.

**Case 1: Text length is insufficient to embed the full watermark.** In practice, when the generated text is too short to encode the full bit string, DERMARK results in failure embedding—some watermark bits are successfully embedded while others fail due to the insufficient text length. In such cases, it is still feasible to increase the watermark strength $\delta$ to achieve successful watermark embedding. However, we should note that a large $\delta$ would significantly compromise text quality. Setting $\delta = 1$ follows standard practice in most related works and represents a well-established trade-off between text quality and watermarking effectiveness. In addition, truncating the watermark is also not a viable solution. Only the generated text is available to the detector. If the watermark length were variable or unknown due to truncation, detection accuracy would drastically decline because the detector would not know how many bits to extract.

Importantly, DERMARK is designed for higher capacity by adaptively allocating bits based on each token's watermark-carrying capability. If our method fails to embed the complete watermark when the text is short, uniform segmentation approaches would fail even more severely.

**Case 2: Text length is significantly longer than the watermark bit length.** Such cases are conceptually inconsistent with the multi-bit watermarking goal, which aims to encode as much identifying information as possible. Embedding only a small number of bits into a very long text underutilizes the available capacity and runs counter to the intent of multi-bit watermarking. Nevertheless, if such a scenario arises, our method can still embed the watermark with minimal semantic disruption, concentrating on embedding within short segments and leaving most of the text untouched.

By contrast, baseline methods such as Balance-Marking distribute the watermark across the entire text to improve robustness. While this approach may enhance resilience to editing, it inevitably increases the overall semantic distortion. In such cases, choosing between the two approaches depends on application priorities: our method favors semantic preservation and high information density, while baselines may be preferred when robustness is paramount and semantic drift is tolerable.

Overall, our method targets a core challenge of multi-bit watermarking: how to maximize watermark density and semantic fidelity in capacity-limited settings. In capacity-abundant scenarios, the trade-offs become more nuanced and context-dependent, with our method and baseline approaches each offering unique advantages.

### F.2 VARIABLE-LENGTH WATERMARK

In the context of this study, we consider that watermarks are embedded into the outputs of different LLMs as user-specific identifiers. Consequently, we assume that multi-bit watermarks are of fixed length $K$. However, in practice, there may be cases where watermarks are of variable length, meaning that the goal is to embed as many watermark bits as possible into the LLM-generated text.

In this context, the advantages of DERMARK can be fully leveraged. The method enables real-time assessment of the generated text to determine whether there is sufficient capacity to embed watermark bits. This dynamic decision-making process ensures that each watermark bit is successfully embedded without significantly affecting the semantics of the text, while simultaneously maximizing the number of watermark bits embedded within the text.

### F.3 EXPLANATION AND SELECTION OF HYPERPARAMETERS

$\alpha$. The parameter $\alpha$ represents the confidence threshold used in the embedding inequality (Eq. (5)), controlling how strictly a segment must satisfy the bit alignment condition. A smaller $\alpha$ leads to more conservative segment acceptance, requiring a higher proportion of aligned tokens. In our experiments, we systematically varied $\alpha$ within the range $[0.8, 0.95]$ to analyze its effect on watermark capacity. For all other experiments, we report results across this range to capture the full performance profile under different confidence levels.

$\delta$. The watermark strength $\delta$ determines the magnitude of the bias applied to the logits of green or red tokens during watermark embedding. A higher $\delta$ increases the separation between token distributions, thus improving embedding robustness and detection reliability. In our capacity and quality experiments, we sweep over a wide range of $\delta$ values to assess its impact. For all other evaluations, we follow the default setting in KGW and fix $\delta = 1$, which provides a good balance between imperceptibility and robustness.

Our watermark extraction relies on the inequality condition in Eq. (5) to identify valid watermark segments. However, since the generated tokens are discrete and sampled probabilistically, the observed token proportions in each segment cannot strictly satisfy the inequality. Even in clean, watermarked texts, a small positive bias is inevitable, as the sampling process introduces stochastic deviation. Furthermore, if the text has undergone insertion or deletion attacks, additional perturbations will lead to unknown deviations in token distributions. To account for these effects, we introduce a bias-correction term $\epsilon_s$ in the extraction loss (cf. Eq. (7)), which tolerates small violations of the inequality caused by stochasticity or editing.

In addition to inequality violations, another source of systematic deviation arises from the fact that each token only has a $(1 - \alpha)$ probability of being sampled from the intended list (e.g., the green list for embedding bit-1). As a result, even under ideal segmentation, each segment may naturally contain an $\alpha$-fraction of mismatched (e.g., red list) tokens. To accommodate this, we introduce a second correction term $\epsilon_c$(cf. Eq. (8)), which adjusts for the expected level of token-type noise under the bit embedding distribution.

Rather than setting $\epsilon_s$ and $\epsilon_c$ manually, we adopt an iterative estimation strategy for both. In each iteration, we first compute the optimal segmentation via dynamic programming. Then, $\epsilon_s$ is updated based on the average residual in Eq. (5) across all segments, while $\epsilon_c$ is updated based on the observed deviation between empirical and expected token-type distributions. This process is repeated until both parameters converge, ensuring robustness to both stochastic and adversarial perturbations.

We also include other hyperparameters to handle estimation stability and loss balancing. The smoothing parameter $\lambda$ is used to stabilize the estimation of token-type priors ($P_{k1}$ and $P_{k0}$) in Eq. (6), particularly in short segments. This parameter is selected through manual tuning on validation data to balance numerical stability and estimation accuracy.

Finally, $\beta$ is a tunable coefficient that controls the relative weight of the two components in Eq. (9), balancing segment-wise fidelity and global bit alignment. After extensive experimentation, we found that setting $\beta$ to 34 yields the best overall performance.

### F.4 PERFORMANCE ON HIGH-ENTROPY DATASETS

To strengthen the rigor of our evaluation, we construct a subset consisting of the top 25% of datasets with the highest watermark detection rates under the baseline, which we refer to as the "high-entropy dataset." As illustrated in Fig. 6a, while DERMARK on `OPT-1.3b` exhibits comparable performance to Balance-Marking, DERMARK on `LLaMA-2-7b` significantly outperforms Balance-Marking. This demonstrates that, as the model size increases, even for texts with high watermark capacity, Balance-Marking is more significantly impacted, whereas DERMARK remains largely unaffected. This observation strongly underscores the practical advantages of DERMARK. In fact, the high-entropy dataset corresponds precisely to the case discussed earlier, where the generated text possesses sufficient capacity to embed multiple watermark bits. Most samples in this subset offer ample entropy, allowing reliable and effective multi-bit watermark embedding.

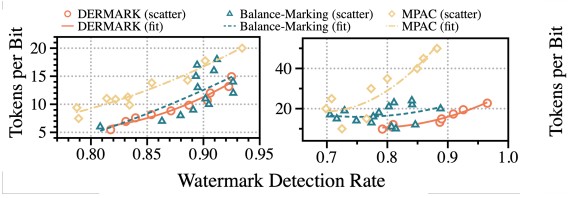
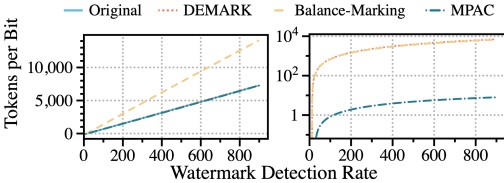

(a) Capacity Comparison across high-entropy dataset on `OPT-1.3b` (left) and `LLaMA-2-7b` (right).

(b) Comparison of time cost in text generation (left) and watermark extraction (right) on `LLaMA-2-70b`.

Figure 6: Supplementary comparison experiments.

### F.5 LARGE-SCALE MODEL DEPLOYMENT

To further demonstrate the practicality of DERMARK on large-scale LLMs, we evaluated the additional overhead on `LLaMA-2-70b` Touvron et al. (2023). The experimental results, as illustrated in Fig. 4b, show that the extra time overhead introduced by DERMARK during the watermark embedding phase is negligible relative to the total inference cost. In contrast, Balance-Marking incurs nearly double the overhead. While this issue may not be as pronounced for smaller LLMs where reasoning is faster, the impact becomes significantly more detrimental as the model scale increases and inference time lengthens. For the watermark extraction phase, the overhead of DERMARK is nearly identical to that of Balance-Marking.

In summary, the additional overhead introduced by DERMARK is negligible across LLMs of varying scales, underscoring its superior practicality.

### F.6 INSTRUCTED MODELS AND LONG-FORM DATASETS

We employed `Llama-3.1-8B-Instruct` as the base model to simulate a realistic chat environment. For the dataset, we utilized the `sentence-transformers/eli5` dataset, which is designed for long-form QA tasks. We randomly sampled 100 queries from the dataset to serve as prompts for generation. Consistent with our main experiments, we set the watermark strength parameter $\delta = 1$. The generation length was fixed at $T = 500/1000$ tokens to evaluate performance in long-context scenarios. We compared our proposed method, DERMARK, against two strong baselines: Balance-Marking and MPAC.

As shown in Figure 7, DERMARK demonstrates significant advantages over the baselines in this instructed generation setting:

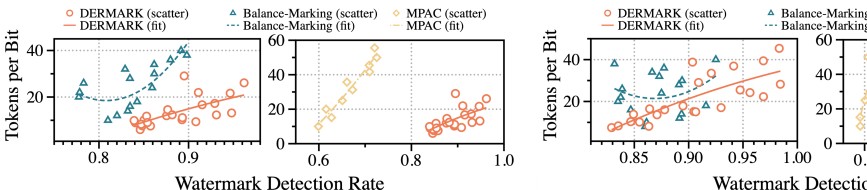

(a) Generation Length = 500. (left) DERMARK vs Balance-Marking, (right) DERMARK vs MPAC.

(b) Generation Length = 1000. (left) DERMARK vs Balance-Marking, (right) DERMARK vs MPAC.

Figure 7: Performance comparison on `Llama-3.1-8B-Instruct` using the ELI5 dataset.

**Comparison with Balance-Marking:** DERMARK consistently outperforms Balance-Marking, particularly in long-text scenarios. At the same detection accuracy level, DERMARK requires significantly fewer tokens to encode a bit of watermark—saving at least **10 tokens per bit** for 500-token texts and **5 tokens per bit** for 1000-token texts. Furthermore, DERMARK exhibits greater stability (lower variance) and easily achieves a detection rate exceeding 95%.

**Comparison with MPAC:** The performance gap is even more pronounced when compared to MPAC. DERMARK achieves substantially higher detection rates while maintaining superior encoding efficiency.

We further evaluated the text quality of the watermarked generation by calculating the Perplexity (PPL) for all three methods. The results indicate that DERMARK, Balance-Marking, and MPAC exhibit similar impacts on PPL. We attribute this similarity to the underlying mechanism shared by these methods: they all embed the watermark by adding a bias $\delta$ to the logits. Since the watermark strength parameter $\delta$ was kept consistent ($\delta = 1$) across all methods in this experiment, the resulting perturbation to the language model's probability distribution remains comparable. This confirms that DERMARK achieves superior efficiency and detectability without incurring an additional penalty on text quality compared to existing methods.

### F.7 Empirical Analysis of CLT Approximation Error

To address potential concerns regarding the reliability of the Central Limit Theorem (CLT) approximation in the short-segment regime (e.g., $N < 20$), we conducted a finite-sample calibration analysis using Monte Carlo simulations. We utilized `Llama-3.1-8B-Instruct` to generate watermarked texts and estimated the true empirical confidence ($P_{True}$) compared to the theoretical confidence predicted by CLT ($P_{CLT}$).The results are presented in Table 3. We grouped the samples by their predicted $P_{CLT}$ and calculated the average empirical accuracy and average segment length for each group.

- Segment Length: The analysis covers segment lengths ranging from approximately $N = 6$ to $N = 26$, directly addressing the low-$N$ regime where asymptotic assumptions are typically most vulnerable.

- Approximation Error: We define the approximation error as $|P_{CLT} - P_{True}|$. As shown in the table, the error remains consistently low across the spectrum. For instance, at a target confidence of 0.850 with an average length of just 6.0 tokens, the deviation is only 0.004. The average error across all tested intervals is approximately 0.026, with a maximum deviation of 0.056.

These empirical results demonstrate that while CLT is an asymptotic result, its approximation in our specific application provides a sufficiently tight bound even for short segments, validating the theoretical foundation of our method in practical settings.

### F.8 Comparision with Provably Robust Watermarking on Llama-2-7b

To provide a comprehensive assessment of robustness, we conducted a direct comparison between DERMARK and provably Qu et al. (2025) (referred to as [1]). The experiments were performed using the `Llama-2-7b` under three types of attacks: random insertion, deletion, and substitution, with noise ratios set at 5% and 10%.

Table 3: Empirical analysis of CLT approximation error and finite-sample correction. We compare the CLT predicted confidence $P_{CLT}$ against the true empirical confidence $P_{True}$ estimated via Monte Carlo simulations across varying segment lengths N on `Llama-3.1-8B-Instruct`.

| $P_{CLT}$ | $P_{True}$ | Segment Length | Approximation Error |
|---|---|---|---|
| 0.850 | 0.846 | 6.000 | 0.004 |
| 0.855 | 0.841 | 8.937 | 0.014 |
| 0.860 | 0.850 | 10.131 | 0.010 |
| 0.865 | 0.848 | 7.644 | 0.017 |
| 0.870 | 0.858 | 7.554 | 0.012 |
| 0.875 | 0.848 | 8.132 | 0.027 |
| 0.880 | 0.847 | 8.303 | 0.033 |
| 0.885 | 0.854 | 9.045 | 0.031 |
| 0.890 | 0.853 | 11.775 | 0.037 |
| 0.895 | 0.839 | 9.919 | 0.056 |
| 0.900 | 0.908 | 9.388 | -0.008 |
| 0.905 | 0.882 | 12.354 | 0.023 |
| 0.910 | 0.893 | 10.251 | 0.017 |
| 0.915 | 0.875 | 11.967 | 0.040 |
| 0.920 | 0.893 | 11.139 | 0.027 |
| 0.925 | 0.877 | 14.581 | 0.048 |
| 0.930 | 0.915 | 16.675 | 0.015 |
| 0.935 | 0.931 | 12.308 | 0.004 |
| 0.940 | 0.911 | 21.947 | 0.029 |
| 0.945 | 0.895 | 29.068 | 0.050 |
| 0.950 | 0.948 | 13.159 | 0.002 |
| 0.955 | 0.929 | 17.249 | 0.026 |
| 0.960 | 0.946 | 21.647 | 0.014 |
| 0.965 | 0.962 | 26.092 | 0.003 |

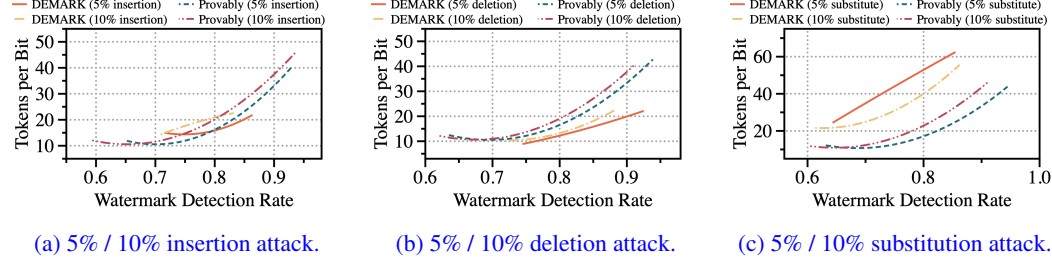

(a) 5% / 10% insertion attack.  (b) 5% / 10% deletion attack.  (c) 5% / 10% substitution attack.

Figure 8: Robustness comparison on `Llama-2-7B` between DERMARK and provably.

The comparative results indicate the following:

**Insertion and Deletion:** DERMARK demonstrates strong resilience against structural perturbations. Under insertion attacks (5% and 10%), DERMARK exhibits robustness levels comparable to [1]. Notably, under deletion attacks, DERMARK slightly outperforms [1], maintaining higher detection accuracy at similar encoding efficiencies.

**Substitution:** In the case of random substitution attacks, we observe that DERMARK's robustness is lower than that of [1].

We attribute the performance gap in substitution attacks to the underlying segmentation mechanisms. The method in [1] employs a non-continuous segmentation strategy, which is inherently more resistant to local token replacements as it avoids dependency on contiguous blocks. In contrast, DERMARK currently utilizes a continuous segmentation strategy. While this approach is effective for structural synchronization (insertion/deletion), it is more sensitive to in-place substitutions that can disrupt the continuous context required for segment identification. We identify this as a limitation of

the current version and plan to incorporate non-continuous segmentation strategies in future work to enhance robustness against substitution attacks.

