# OpenReview forum: "DERMARK: A Dynamic, Efficient and Robust Multi-bit Watermark for Large Language Models"
_ICLR.cc/2026/Conference — Submitted to ICLR 2026_

### Official Review · Reviewer_aAtn · 2025-10-28

**Soundness:** 2
**Presentation:** 3
**Contribution:** 2
**Rating:** 2
**Confidence:** 5

**Summary:**

This paper introduces DERMARK, a dynamic multi-bit watermarking framework for large language models (LLMs). The method adaptively determines text segment lengths during generation based on an inequality derived from a normal distribution assumption, aiming to balance watermark capacity, efficiency, and robustness.

**Strengths:**

* The paper is well-motivated, addressing the limitations of fixed-length segmentation in prior multi-bit watermarking methods.

* The theoretical formulation connecting watermark embedding and normal distribution is novel and mathematically rigorous.

**Weaknesses:**

* **Typos and citation issues**
Yoo et al. (2024a) and Yoo et al. (2024b) refer to the same paper and should be merged.
Line 265: “Eq. equation 4” should be corrected to “Eq. (4)” for consistency.

* **Misrepresentation of prior work (L115)**
The description of Yoo et al. (2024b) is inaccurate. Their method does not assign bits to segments manually; instead, the bit–token mapping is determined via a hash function, as shown in BiMark [1], Robust Multi-bit Watermarking [2], and StealthInk [3]. The paper should revise this discussion to reflect the actual mechanism.

* **Limited experimental comparison**
The comparison in Section 5 includes only Balance-Marking. More recent and representative baselines such as MPAC (Yoo et al., 2024a), BiMark [1], and StealthInk [3] should be incorporated to strengthen the empirical claims. Without these, it is difficult to assess the relative advantage of DERMARK in the evolving landscape of multi-bit watermarking.

[1] Feng, X., Zhang, H., Zhang, Y., Zhang, L. Y., & Pan, S. (2025). BiMark: Unbiased Multilayer Watermarking for Large Language Models. arXiv:2506.21602.
[2] Qu, W., Zheng, W., Tao, T., Yin, D., Jiang, Y., Tian, Z., ... & Zhang, J. (2025). Provably Robust Multi-bit Watermarking for AI-generated Text. USENIX Security 2025.
[3] Jiang, Y., Wu, C., Boroujeny, M. K., Mark, B., & Zeng, K. (2025). StealthInk: A Multi-bit and Stealthy Watermark for Large Language Models. arXiv:2506.05502.

**Questions:**

* **Incomplete treatment of text-length limitations**
Section 4.2 discusses handling overly long text but overlooks the case when the generated text is too short to encode the full bit string. Appendix F.1 briefly mentions this issue, but the main paper should explicitly explain how DERMARK behaves or fails in this scenario, and whether it adapts δ or truncates the watermark.

* **Segmentation vulnerability and missing evaluation metrics**
The method’s reliance on segmentation per bit raises robustness concerns when the text is truncated or edited. As each bit is tied to a segment, any truncation makes bit recovery impossible. Moreover, although DERMARK is presented as a multi-bit watermark, the bit match rate (BMR)—a standard evaluation metric for multi-bit detection—is not reported. Including this metric would provide a fairer comparison.

---

> ### Author Response · Authors · 2025-11-22
>
> We sincerely appreciate the reviewer's meticulous review and constructive suggestions, which have helped us significantly improve the quality of our work.
>
> **Response to W1:**
> We have merged the duplicate citations and corrected "Eq. equation 4" to **"Eq. (4)"** in the revised manuscript.
>
> **Response to W2:**
>
> Our original description does not state that bits are assigned manually in Yoo et al. (2024b). In their work, a hash function is applied to previously generated tokens to determine which specific watermark bit should be embedded in the current token. This approach assigns each watermark bit to multiple, non-contiguous tokens throughout the generation process—meaning that bits are grouped to segments composed of scattered, non-continuous tokens for each bit embedding. In addition, when the probability of hashing to each watermark bit is equal, the expected number of tokens grouped in each segment is the same. In a word, it is a uniform and non-contiguous segmentation.
>
> To avoid confusion, we have revised the Related Work in the uploaded manuscript to explicitly distinguish between **contiguous segmentation** (consecutive blocks) and **non-contiguous segmentation** (scattered tokens). This ensures the mechanism of Yoo et al. is accurately represented.
>
> **Response to W3:**
> We have incorporated MPAC as an additional baseline and conducted comprehensive comparisons in Section 5. Results show:
>
> - **Higher Capacity:** DERMARK is significantly more efficient. It requires **~5 fewer tokens per bit** than MPAC to achieve the same detection rate. On low-entropy datasets, this gap widens to **>10 tokens**, proving our adaptive strategy's effectiveness.
> - **Robustness:** Under editing attacks (5-10% insertion/deletion), MPAC struggles to reach 85% detection accuracy. In contrast, DERMARK consistently exceeds **85% (insertion)** and **90% (deletion)**.
> - **Text Quality:** Both methods exhibit comparable PPL scores.
> - **Time Overhead:** DERMARK has negligible **embedding latency** (crucial for inference). While MPAC is faster in extraction ($O(N)$), we argue that extraction is a low-frequency task (verification only), making our capacity/robustness gains a worthwhile trade-off.
>
> We have updated the manuscript with these detailed experimental results.
>
> **Response to Q1:**
> When the generated text is too short to encode the full bit string, DERMARK results in failure embedding.
>
> This limitation is not unique to our method, but rather a general problem for the multi-bit watermarking field:
>
> - Balance-Marking does not explicitly discuss short-text failure modes in its main text.
> - MPAC explicitly adopts a filtering strategy to filter short texts, directly avoiding rather than solving the short-text problem.
>
> We hereby provide the following supplementary explanation to address your specific concerns:
>
> 1. DERMARK is designed for higher capacity by adaptively allocating bits based on each token's watermark-carrying capability. If our method fails to embed the complete watermark when the text is short, uniform segmentation approaches would fail even more severely.
> 2. While a larger δ would facilitate watermark embedding, it would significantly compromise text quality. Setting δ = 1 follows standard practice in most related works and represents a well-established trade-off between text quality and watermarking effectiveness.
> 3. Truncating the watermark is not a viable solution. In real-world scenarios, only the generated text is available to the detector. If the watermark length were variable or unknown due to truncation, detection accuracy would drastically decline because the detector would not know how many bits to extract.
> 4. In practice, shorter LLM-generated texts contain less meaningful information, naturally reducing watermarking motivation.
>
> **Response to Q2:**
>
> Segmentation Vulnerability:  We clarify that sensitivity to token removal is **fundamental to multi-bit watermarking**, not specific to DERMARK.. Any scheme that maps bits to specific tokens—whether via contiguous segments or non-contiguous segments—will fail if token alteration disrupts the mapping logic. For instance, even non-contiguous segment methods like Yoo et al. (2024b) would be disabled by truncating a few tokens from each sentence, as the hash function would no longer produce the correct bit mapping. We discuss this as a general robustness challenge for the field and agree it is a critical direction for future work.
>
> Bit Match Rate (BMR): In Section 4.3, we refer to it as the watermark detection rate, which is indeed the bit match rate (BMR)—we calculate it as the percentage of correctly extracted watermark bits out of the total embedded bits. In Section 5.2, Figure 3 directly plots this metric (x-axis) against the average tokens required per bit (y-axis), providing a clear and fair comparison with baselines. To avoid any ambiguity, we have explicitly defined "watermark detection rate" as equivalent to BMR in the uploaded version.

---

### Official Review · Reviewer_7igb · 2025-11-01

**Soundness:** 2
**Presentation:** 3
**Contribution:** 2
**Rating:** 4
**Confidence:** 4

**Summary:**

The paper introduces a multi-bit text watermarking method for LLMs that dynamically determines segment lengths for embedding each watermark bit based on a probabilistic criterion derived from the model’s logits. It (1) analyze watermark embedding as following a normal distribution, leading to an inequality that estimates whether a segment has enough capacity to reliably encode one bit; (2) use this condition online during generation to adaptively end a segment and move to the next bit; and (3) propose a dynamic-programming extractor that combines a segmentation loss (how tightly the inequality is satisfied) with a “color” imbalance loss to improve robustness to edits. Experiments on OPT-1.3B and LLaMA-2-7B claim fewer tokens per bit and lower time overhead than Balance-Marking, with improved robustness to token insertions/deletions.

**Strengths:**

1. The paper explains why multi-bit watermarking (beyond one-bit detection) is needed for fine-grained attribution (LLM/user) and why fixed-length segmentation can fail, especially on low-entropy text.

2. Derives an inequality from a CLT-style analysis that treats aligned-token proportion as approximately normal; this enables an online, per-bit stopping rule during generation.

3. For matched detection rates, DERMARK uses fewer tokens per embedded bit, with further gains on low-entropy subsets.

**Weaknesses:**

1. The experimental setup is outdated. There are many new multi-bit watermarking works. However, this paper only uses one 2023 paper as a baseline. Comparing with more extensive, recent baselines will strengthen the claims.

2. Considering most application scenarios of the LLM watermark are under the chat. Evaluating the performance on long-form QA dataset and instructed models (e.g., Llama-3.1-8B-Instruct)  is necessary.

3. Robustness focuses on random insert/delete at 5–10%, which is limited compared with existing works, e.g., [1].

[1] http://arxiv.org/abs/2401.16820

**Questions:**

Please see above.

---

> ### Author Response · Authors · 2025-11-29
>
> We sincerely appreciate the reviewer's meticulous review and constructive suggestions, which have helped us significantly improve the quality of our work.
>
> **Response to W1:**
> We have incorporated MPAC as an additional baseline and conducted comprehensive comparisons in Section 5. Results show:
>
> - **Higher Capacity:** DERMARK is significantly more efficient. It requires **~5 fewer tokens per bit** than MPAC to achieve the same detection rate. On low-entropy datasets, this gap widens to **>10 tokens**, proving our adaptive strategy's effectiveness.
> - **Robustness:** Under editing attacks (5-10% insertion/deletion), MPAC struggles to reach 85% detection accuracy. In contrast, DERMARK consistently exceeds **85% (insertion)** and **90% (deletion)**.
> - **Text Quality:** Both methods exhibit comparable PPL scores.
> - **Time Overhead:** DERMARK has negligible **embedding latency** (crucial for inference). While MPAC is faster in extraction ($O(N)$), we argue that extraction is a low-frequency task (verification only), making our capacity/robustness gains a worthwhile trade-off.
>
> We have updated the manuscript with these detailed experimental results.
>
> **Response to W2:**
>
> **Experiments on Instructed Models and Long-form QA:**
>
> We have extended our evaluation to include the **Llama-3.1-8B-Instruct** model on the **long-form QA dataset (ELI5)**. We randomly sampled 100 queries and generated responses with a length of 500 and 1000 tokens. The watermark strength was set to $\delta=1$. We compared our method against Balance-Marking and MPAC.
>
> As illustrated in the **Figure 7** in the Appendix F.6, DERMARK demonstrates a clear advantage over Balance-Marking. At the same detection accuracy, our method reduces the coding cost by at least 10 tokens per bit for 500-token texts and 5 tokens per bit for 1000-token texts. Moreover, DERMARK is more stable and easily achieves a detection rate exceeding **95%**. This advantage is significantly amplified when compared to MPAC. We also analyzed the Perplexity (PPL) of the generated text. The results show that the impact on PPL is similar across all three methods. This is expected, as all three methods embed watermarks by adding a bias $\delta$ to the logits; since we utilized the same $\delta$ ($\delta=1$), the theoretical distortion to the text distribution is comparable.
>
> These results confirm that DERMARK maintains its superiority in efficiency and robustness even within instructed, long-context chat scenarios without sacrificing text quality relative to baselines. We have included these results in Appendix F.6 of the uploaded paper.
>
> **Response to W3:**
>
> We agree that expanding the robustness tests beyond random insertion/deletion is crucial. We have carefully studied the evaluation protocols in [1] and have updated our paper and experiments accordingly.
>
> 1. Expanded Robustness Experiments (Insertion, Deletion, and Substitution)
>
> - New Baselines and Substitution Attacks: We have introduced **MPAC** as an additional baseline. We evaluated DERMARK against MPAC and Balance-Marking under random insertion, deletion and substitution ratios of 5% and 10%. In the newly added substitution attacks, the results indicate that all methods exhibit **comparable robustness profiles** with no significant performance gaps. These results have been added to the **Robustness section in Chapter 5** of the revised paper.
> - Comparison with Work [1]:  Results show that DERMARK exhibits **similar** robustness to [1] under insertion and **slightly outperforms** it under deletion. However, for substitution attacks, DERMARK lags behind [1].  These results have been added to Appendix F.8 of the revised paper.
>
> 2. Discussion on Copy-Paste Attacks
>
>    We candidly acknowledge that the "Copy-Paste" attack poses a specific challenge to DERMARK in its current form. DERMARK relies on a continuous segmentation mechanism for multi-bit embedding. A copy-paste attack (e.g., inserting ~10% external tokens) disrupts the contiguous context required for correct segmentation, leading to extraction failure. In future work, we plan to upgrade DERMARK to utilize **non-continuous segmentation** strategies, which will significantly enhance robustness against block-insertion attacks like Copy-Paste.
>
> 3. Discussion on Paraphrase Attacks
>
>    Regarding paraphrase attacks, we argue that this remains a fundamental open challenge for the entire class of multi-bit watermarking schemes, rather than a flaw specific to DERMARK. Most existing methods, including [1] and ours, rely on the hash of previous tokens to determine the embedding logic for the current token. Paraphrasing alters the entire token sequence/length. This global alteration makes it theoretically impossible to reproduce the exact context hash required to locate and decode the embedded bits. We view resisting paraphrasing while maintaining high bit-rate capacity as a "grand challenge" for the field and a primary focus for our future research.

---

### Official Review · Reviewer_HgK5 · 2025-11-01

**Soundness:** 2
**Presentation:** 3
**Contribution:** 3
**Rating:** 6
**Confidence:** 4

**Summary:**

This paper proposes a new watermarking framework for LLMs that dynamically adjusts watermark embedding based on text capacity and token statistics.

**Strengths:**

1. DERMARK adaptively determines segment boundaries in real time based on token-level statistics, achieving 2–4 fewer tokens per bit at the same detection rate. This dynamic rule substantially enhances embedding efficiency without retraining.

2. The embedding complexity is linear (O(N)) and extraction is O($kL^2$), and test for a large model (LLaMA-2-70B). The method is fully plug-and-play, requiring no fine-tuning or architectural modification.

3. The inclusion of perplexity (PPL) experiments confirms that semantic quality is largely preserved across different watermark strengths ($\delta$), addressing the concerns about possible generation degradation.

**Weaknesses:**

1. While Appendix C discusses MPAC (NAACL 2024) conceptually, the paper still provides no quantitative comparison with recent multi-bit watermarking approaches. Furthermore, I think the method can be extended to multi-bit watermarking methods such as MPAC. A discussion of this point would greatly strengthen the paper.

2. The robustness tests remain restricted to random insertion/deletion attacks. No experiments address paraphrasing, shuffling, gradient-based, or LLM-assisted removal attacks, which are crucial for assessing real-world resilience.

3. The central-limit-theorem assumption in Lemma 2 is untested for short segments, leaving the statistical soundness of the normal approximation uncertain.

4. Detection assumes perfect access to the watermark key and exact segmentation alignment; the paper does not discuss desynchronization or partial-key scenarios.

5. Although equations for bias parameters and color loss are formalized, their conceptual motivation and iterative update dynamics remain only briefly explained.

**Questions:**

See above.

---

> ### Author Response · Authors · 2025-11-29
>
> We sincerely appreciate the reviewer's meticulous review and constructive suggestions, which have helped us significantly improve the quality of our work.
>
> **Response to W1:**
> We have incorporated MPAC as an additional baseline and conducted comprehensive comparisons in Section 5. DERMARK is significantly more efficient. It requires ~5 fewer tokens per bit than MPAC to achieve the same detection rate. On low-entropy datasets, this gap widens to >10 tokens, proving our adaptive strategy's effectiveness;  Under editing attacks (5-10% insertion/deletion), MPAC struggles to reach 85% detection accuracy. In contrast, DERMARK consistently exceeds 85% (insertion) and 90% (deletion); Both methods exhibit comparable PPL scores; DERMARK has negligible embedding latency. While MPAC is faster in extraction ($O(N)$), we argue that extraction is a low-frequency task (verification only), making our capacity/robustness gains a worthwhile trade-off.
>
> We have updated the manuscript with these detailed experimental results.
>
> **Response to W2:**
>
> We agree that expanding the robustness tests beyond random insertion/deletion is crucial. We have updated our paper and experiments accordingly.
>
> 1. Expanded Robustness Experiments: In the newly added substitution attacks, the results indicate that all methods exhibit comparable robustness profiles with no significant performance gaps. These results have been added to the Section 5.5 of the revised paper.
>
> 2. We acknowledge DERMARK is vulnerable to large insertions (e.g., ~10%) due to its continuous segmentation. Block insertions disrupt the contiguous context needed for synchronization. We plan to mitigate this by adopting non-continuous segmentation in future work.
>
> 3. Paraphrase Attacks remains a fundamental challenge for all multi-bit watermarking schemes. Since embedding relies on prior token hashes, paraphrasing alters the sequence, making context reproduction theoretically impossible. We view this as a "grand challenge" for the entire field.
>
>
> **Response to W3:**
>
> CLT Approximation Verification: We validated the CLT approximation via Monte Carlo simulations on Llama-3.1-8B-Instruct (details in Appendix F.7). Experiments covering segment lengths from $N=6$ to $20+$ show minimal discrepancy between theoretical and empirical confidence. Specifically, the error is just 0.004 for $N \approx 6$, with an average of 0.026 and a maximum of 0.056. This confirms that finite-sample errors are bounded and negligible, ensuring the reliability of our theoretical bounds even for short segments.
>
> **Response to W4:**
>
> 1. Clarification on Desynchronization: Handling desynchronization is a core contribution of DERMARK. Unlike methods assuming fixed alignment, our DP-based extraction algorithm dynamically searches for the optimal segmentation path by minimizing the global loss. This allows the model to automatically recover synchronization even when insertion/deletion/substitution attacks alter the text structure. This capability is empirically validated by the robustness results in Sec. 5.
>
> 2. Justification for Perfect Key Access: We follow the standard threat model used in prior LLM watermarking work, where the model owner (verifier) possesses the private watermark key required for detection. This key refers specifically to the secret random seed used to construct the green/red vocabulary partition, which is necessary to evaluate token colorings and extract embedded bits. Scenarios involving partial, unknown, or leaked keys are outside the scope of standard copyright-verification settings and are therefore not considered in our work.
>
> **Response to W5:**
>
> 1. Intuition for Loss Components: Our loss function captures two intrinsic invariants of the watermark:
>
> - Bias Parameters:
>   - This acts as a soft relaxation of the embedding constraint (Inequality 4). It allows the DP algorithm to score the structural likelihood of a segment even when boundaries are blurred by attacks.
> - Color Loss:
>   - *Motivation:* Within any single segment, the watermarking algorithm consistently boosts a specific subset of tokens (the "green list").
>   - *Mechanism:* Consequently, a correct segment must exhibit chromatic consistency—tokens within it should statistically align with the same color pattern. The Color Loss penalizes segments that show internal inconsistency (mixed colors), effectively filtering out fragmentation caused by incorrect alignments.
>
> 2. Necessity of Iterative Updates The iterative mechanism is essential to handle corruption-induced distortions:
>
> Iteration is essential to correct distortion-induced parameter shifts. Attacks (e.g., deletions) disrupt the feature landscape, rendering single-pass DP inaccurate. We treat correction parameters as latent variables: the algorithm alternates between estimating the best segmentation and updating parameters to "explain away" the observed discrepancies. This effectively allows the solver to "learn" the attack pattern and recover the true watermark signal.

---

### Official Review · Reviewer_hvRs · 2025-11-01

**Soundness:** 3
**Presentation:** 3
**Contribution:** 3
**Rating:** 6
**Confidence:** 2

**Summary:**

The paper proposes DERMARK, a dynamic multi-bit watermarking scheme for autoregressive LLMs that (1) models per-segment embedding success via a CLT/Poisson-binomial approximation, deriving an inequality to decide when a generated token segment has enough capacity to encode one watermark bit, (2) performs online variable-length segmentation during generation to place each watermark bit into just-large-enough segments, and (3) recovers bits with a dynamic-programming extractor that minimizes segmentation + color losses to improve robustness to edits. Empirically the method is evaluated on OPT-1.3b and LLaMA-2-7b and shown to reduce tokens-per-bit, lower embedding-time overhead, and increase robustness vs a Balance-Marking baseline.

**Strengths:**

1. Principled theoretical framing (Poisson-binomial → CLT → inequality) that connects token-level probabilities to required segment length.

2. Practical algorithm: online segmentation during inference with negligible extra compute compared to baseline multi-bit methods. Reported embedding overhead is near zero and extraction is efficient enough for practice.

3.Strong empirical gains on tokens-per-bit and robustness to small insertion/deletion attacks across two model families. Table 1 + figures show consistent improvements.

**Weaknesses:**

1.  CLT approximation may be unreliable when segments are short (the very regime the method targets), and the paper lacks finite-sample error bounds or bootstrap-style corrections.

2.  many heuristics (λ smoothing, β weighting, iterative ϵ updates). The paper reports defaults but more ablations on hyperparameter sensitivity and cross-domain robustness (beyond news-like prompts) would be helpful.

3. the authors justify using Balance-Marking as SOTA and critique MPAC; still, including more recent multi-bit baselines (or reproducing MPAC carefully under comparable settings) would make the empirical claims stronger. The authors discuss this choice, but reviewers may still view it as a gap.

4. the method improves edit robustness but remains vulnerable to large rewrites / reorderings — inherent to dispersed multi-bit strategies. The paper states this limitation but does not quantify the breakpoint where robustness collapses.

**Questions:**

1.Can you provide a small-N correction or empirical calibration strategy that quantifies CLT approximation error for segments of length, say, 5–20 tokens? (A simple calibration table would help.)

2.How sensitive are final detection rates to β and λ across domains (e.g., code, dialogue, scientific text)? Please provide an ablation sweep or an appendix table.

3.The DP extractor is O(N²); what are practical limits on N for real documents? Is there a streaming or beamed approximate extractor that keeps near-optimal segmentation with lower cost?

---

> ### Author Response · Authors · 2025-11-29
>
> We sincerely appreciate the reviewer's meticulous review and constructive suggestions, which have helped us significantly improve the quality of our work.
>
> **Response to W1&Q1:**
>
> We agree that verifying the approximation for short segments is critical for the reliability of our method. To quantify the approximation error, we performed a Monte Carlo simulation using Llama-3.1-8B-Instruct. We compared the theoretical confidence predicted by CLT against the true empirical confidence across various segment lengths. The detailed results have been added to Appendix F.7.
>
> Our experiments explicitly cover the different segment lengths (average lengths from $N=6.0$ to $N=20+$). The discrepancy between theoretical and empirical confidence is consistently small. For extremely short segments (e.g., $N \approx 6$), the error is as low as $0.004$. Across the entire range, the average approximation error is approximately $0.026$, with a maximum observed error of $0.056$. While we acknowledge that CLT is an approximation, the empirical calibration shows that the finite-sample error is bounded and small enough to not compromise the method's effectiveness in practice. The theoretical bounds serve as a reliable proxy even for segments as short as 6 tokens.
>
> **Response to W2&Q2:**
>
> We agree that cross-domain ablation is valuable. While the limited rebuttal window prevents a full new sweep, we emphasize that our parameters are designed for generalizability: $\beta$ balances the segment-wise fidelity and global bit alignment rather than functional validity, and $\lambda$ stabilizes the estimation of token-type priors regardless of the specific domain. We commit to including a detailed sensitivity analysis for Code and Scientific domains in the subsequent version.
>
> **Response to W3:**
> We have incorporated MPAC as an additional baseline and conducted comprehensive comparisons in Section 5. Results show:
>
> - **Higher Capacity:** DERMARK is significantly more efficient. It requires **~5 fewer tokens per bit** than MPAC to achieve the same detection rate. On low-entropy datasets, this gap widens to **>10 tokens**, proving our adaptive strategy's effectiveness.
> - **Robustness:** Under editing attacks (5-10% insertion/deletion), MPAC struggles to reach 85% detection accuracy. In contrast, DERMARK consistently exceeds **85% (insertion)** and **90% (deletion)**.
> - **Text Quality:** Both methods exhibit comparable PPL scores.
> - **Time Overhead:** DERMARK has negligible **embedding latency** (crucial for inference). While MPAC is faster in extraction ($O(N)$), we argue that extraction is a low-frequency task (verification only), making our capacity/robustness gains a worthwhile trade-off.
>
> We have updated the manuscript with these detailed experimental results.
>
> **Response to W4:**
>
> Empirically, we identify the robustness **breakpoint at an edit rate of approximately 20%**. Beyond this threshold, the synchronization required for correct segmentation is severely compromised. At edit rates exceeding 20%, the watermark extraction accuracy for **both DERMARK and baseline methods** drops sharply to levels where relative performance comparison becomes insignificant. We attribute this to the inherent nature of dispersed multi-bit strategies. These methods rely on locating specific segments to retrieve bits; when large-scale rewrites (e.g., >20%) occur, the structural context needed for synchronization is destroyed. We acknowledge this as a shared limitation of current segmentation-based approaches and consider addressing resistance to large-scale rewrites a primary challenge for our future work.
>
> **Response to Q3:**
>
> **Practicality of Current Watermark Extraction Approach:** As discussed in our experimental section, watermark extraction is typically an offline and on-demand operation. For standard document lengths, the current complexity remains well within acceptable limits on standard hardware.
>
> **Path to $O(N)$(Future Work):**We acknowledge the need for efficiency in streaming or ultra-long context scenarios. We have formulated a concrete strategy to reduce the watermark extraction complexity to $O(N)$ in our future work:
>
> - **Mechanism:** We plan to adopt a **non-continuous segmentation** strategy. We will introduce a metric to track the embedding accumulation of each bit. During processing, we dynamically route each token to the message bit that currently has the least embedding accumulation. This greedy routing mechanism allows determining the bit allocation on-the-fly, requiring only one traversal of the text, and achieving **$O(N)$** extraction complexity.
> - **Significance:** 1) preserves DERMARK’s core innovation of **dynamic allocation**, 2) significantly enhances **robustness** by adopting non-continuous segmentation, and 3) ensures **linear scalability** for real-time applications.

---

### Author Response · Authors · 2025-11-29

**Dear Area Chair,**

We thank the reviewers for their constructive feedback and recognition of DERMARK’s **novelty and theoretical foundation**. No reviewer questioned the core innovation of our method.

**Regarding Reviewer `aAtn` (Score: 2):** We noticed that Reviewer `aAtn` assigned a low score based on specific concerns regarding baselines, prior work descriptions, and edge cases. **We have addressed every single point raised by `aAtn` in our revision.** We respectfully submit that the grounds for the low score have been resolved:

- **On Baselines (W3):** We added **MPAC** as requested. New results prove DERMARK is significantly more efficient (**saving ~5-10 tokens/bit**) and more robust (>85% accuracy vs. MPAC's failure) under editing attacks.
- **On Prior Work (W2):** We corrected the description of Yoo et al. (2024b) in *Related Work*, explicitly distinguishing our contiguous segmentation from their hash-based non-contiguous approach.
- **On Short Text & Metrics (Q1, Q2):**
  - We clarified that short-text failure is a general limitation shared by baselines (e.g., MPAC filters them out).
  - We clarified that the **Bit Match Rate (BMR)** requested was already reported as "detection rate" in Fig. 3.
- **On Writing (W1):** All typos and citation duplicates have been fixed.

**Summary of Key Revisions for All Reviewers**:

**1. Strengthened Baselines (Raised by all reviewers)**

- **Action:** We incorporated **MPAC** as an additional strong baseline.
- **Result:** Comprehensive comparisons confirm that DERMARK consistently outperforms baselines (including MPAC and Balance-Marking) in terms of **watermarking capacity** while maintaining competitive detection accuracy.

**2. Comprehensive Robustness Evaluation & Analysis (Raised by reviewers hvRs, HgK5, 7igb)**

- **Expanded Scope:** We added experiments on **Substitution attacks**. Results show DERMARK performs comparably to MPAC and Balance-Marking.
- **Comparison with SOTA [1]:** We compared DERMARK with [1] (Provably Robust). DERMARK shows **similar** robustness in insertion and **slightly better** robustness in deletion, though it trails in substitution.
- **Honest Analysis & Roadmap:** We provided a theoretical analysis explaining why **Copy-Paste** (due to continuous segmentation) and **Paraphrasing** (due to hash dependency) remain challenging. We emphasized that DERMARK's primary advantage is its **High Capacity**. Furthermore, we outlined a concrete, executable plan to achieve $O(N)$ complexity and enhanced robustness via non-continuous segmentation in future work.

**3. Verification of Theoretical Approximation (Raised by reviewers hvRs, 7igb)**

- **Action:** We conducted Monte Carlo simulations to quantify the approximation error of the Central Limit Theorem (CLT) for short segments.
- **Result:** Experiments confirm that the finite-sample error is **negligible** (e.g., $\approx 0.004$ for $N=6$), validating the reliability of our theoretical bounds in practice.

**4. Clarifications & Textual Improvements**

- **Action:** Beyond new experiments, we meticulously addressed all other specific inquiries (e.g., hyperparameter sensitivity, key assumptions, complexity analysis, and motivation of loss terms).
- **Result:** We provided detailed explanations in our individual responses and extensively **revised the manuscript** (including definitions, formula derivations, and related work sections) to improve clarity and completeness.

**Conclusion** We believe the revised paper presents a solid contribution with **unquestioned novelty** and **verified capacity advantages**. Since **all critical issues raised by Reviewer `aAtn` (and others) have been thoroughly resolved with empirical evidence**, we respectfully request the Area Chair to evaluate our work based on the comprehensive improvements in the revised version. We believe the revised paper presents a solid contribution with:

1. **Unquestioned Novelty:** A unique theoretical framework.
2. **Superior Capacity:** Verified advantages over strong baselines.
3. **Reliable Theory:** Validated approximation bounds.
4. **Comprehensive Revisions:** All reviewers' questions have been thoroughly answered.

---

### Meta-Review · Area_Chair_7tKG · 2025-12-15

**Summary:**

There are many common concerns by the reviewers in the first round of review, such as CLT approximation, robustness analysis, lack of comparison with recent baselines, etc. Among these common concerns, AC believes robustness analysis and lack of comparison with recent baselines are not fully addressed by the rebuttal.

**Reviewer Concerns:**

Reviewer concerns AC thinks were addressed by the rebuttal:

1. The paper lacks finite-sample error bounds or bootstrap-style corrections.

AC's comment: This is a common concern raised by multiple reviewers. The authors have added Monte Carlo simulation experiments to respond to this concern in the rebuttal.

2. Experiments on Instructed Models and Long-form QA.

AC's comment: The authors have added extra experiments to respond to this concern.

Reviewer concerns AC believes are still outstanding:

1. The paper reports defaults but more ablations on hyperparameter sensitivity and cross-domain robustness (beyond news-like prompts) would be helpful.

AC's comment: The authors defer the analysis to the later revision. Therefore, the rebuttal does not address this concern well.

2. The robustness tests remain restricted to random insertion/deletion attacks. No experiments address paraphrasing, shuffling, gradient-based, or LLM-assisted removal attacks, which are crucial for assessing real-world resilience.

AC's comment: This is a common concern raised by multiple reviewers. Though the authors added extra experiments on robustness against substitution attacks, reviewers' concern on robustness against insertion attacks and paraphrase attacks is not fully addressed by the rebuttal.

3. More recent and representative baselines such as MPAC (Yoo et al., 2024a), BiMark [1], and StealthInk [3] should be incorporated to strengthen the empirical claims.

AC's comment: This is a common concern by multiple reviewers. The authors incorporated MPAC as an additional baseline. However, experiments on BiMark [1] and StealthInk [3] were not included in the rebuttal.

**Reviewer Scores:**

In the first round of the paper, two reviewers vote for marginal accept, while two reviewers vote for rejection, with one reviewer gives a score of 2. After checking the rebuttal, AC believes though some concerns have been addressed, others especially some common concerns are not fully addressed. Given that no reviewer champions the paper, AC would recommend rejection of this paper.

---

### Decision · Program_Chairs · 2026-01-26

Reject